# Combining split-sample testing and Hidden Markov Modeling to assess the robustness of hydrological models

Etienne Guilpart [1], Vahid Espanmanesh [1], Amaury Tilmant [1], and François Anctil [1]

[1]Département de génie civil et de génie des eaux, Université Laval, Québec, Canada

**Correspondence:** Etienne Guilpart (etienne.guilpart@gmail.com)

**Abstract.** The impacts of climate and land-use changes make the stationary assumption in hydrology obsolete. Moreover, there is still considerable uncertainty regarding the future evolution of the Earth's climate and the extent of the alteration of flow regimes. Climate change impact assessment in the water sector typically involves a modelling chain in which a hydrological model is needed to generate hydrologic projections from climate forcings. Considering the inherent uncertainty of the future climate, it is crucial to assess the performance of the hydrologic model over a wide range of climates and their corresponding hydrologic conditions. In this paper, numerous, contrasted, climate sequences identified by a Hidden Markov Model (HMM) are used in a differential split-sample testing framework to assess the robustness of a hydrologic model. The differential split-sample test based on an HMM classification is implemented on the time series of monthly river discharges in the upper Senegal River Basin in West Africa, a region characterized by the presence of low-frequency climate signals. A comparison with the results obtained using classical rupture tests shows that the diversity of hydrologic sequences identified using the HMM can help assessing the robustness of the hydrologic model.

## 1 Introduction

According to some authors, humanity has entered a new geological Epoch, the Anthropocene, characterized by rapid environmental changes due to human activities (Falkenmark et al., 2019). Among those activities, the massive release of carbon dioxide since the industrial revolution is expected to lead to global warming, which in turn will affect the hydrological cycle (Gleeson et al., 2020). In the past, water engineers were able to design and operate water infrastructure based on the assumption that the climate was stationary, and hence that time series of recorded hydrologic variables such as precipitation and river discharge were representative of future hydrologic conditions (Bernier, 1977; Payrastre, 2003; Naghettini, 2017). Now that the climate is changing, this assumption of stationarity is considered obsolete or even "dead" according to Milly et al. (2008). To deal with this issue, water planners and managers have devoted significant efforts to the development of new decision analytic frameworks that explicitly capture the uncertainties attached to climate change and its impacts on water resources (Brown and Wilby, 2012; Prudhomme et al., 2010).

There are essentially two categories of decision-analytic frameworks : top-down versus bottom-up. The first relies on the sequential coupling of models: GCM models are run to project future precipitations and temperatures which are then downscaled and used as inputs to hydrologic models whose outputs are then processed by water systems models (Peel and Blöschl, 2011). This is consistent with the traditional "predict-then-act" decision-making paradigm (Weaver et al., 2013). The second category rather seeks to identify robust solutions, i.e. solutions that will perform relatively well across a wide range of hydrologic conditions (Lempert et al., 2006). In terms of decision-making paradigm, the idea here is to "minimize regret".

Despite their differences, both frameworks rely at some point on a hydrological model to transform the climate forcings into streamflows. The hydrological model can be stochastic (Borgomeo et al., 2014; Poff et al., 2016), distributed or conceptual (Fortin et al., 2007; Ludwig et al., 2009). When the model is conceptual, its performances must be assessed over contrasting climatic periods because it should be able to perform well over contrasted hydro-climatic conditions (Klemes, 1986). For that purpose, the differential split-sample test principle of Klemes (1986) suggests dividing the whole period into independent periods with different stationary features. The hydrological model is then calibrated on a specific period and validated on other(s). However, as the technique used to subdivide a time series affects the intrinsic variability embedded in the subsequences, it may impact the calibration and validation steps (Thirel et al., 2015a,b; Stephens et al., 2019; Motavita et al., 2019; Dakhlaoui et al., 2019; Huang et al., 2020).

Several statistical tests have then been proposed to detect shifts and trends in time series including the Mann-Kendall test (Mann, 1945; Kendall, 1948) and the Pettitt test (Pettitt, 1979). A review of those tests can be found in Liu et al. (2016). However, most of those tests can only make the distinction between two periods, before and after the change point, and are therefore unable to handle more complex climate sequences with multiple change points. In certain regions, for example, time series of river discharges are characterized by low-frequency shifts, and hence multiple change points, because the underlying hydrological processes are influenced by low-frequency climate signals such as El Nino Southern Oscillation (Bracken et al., 2014; Nalley et al., 2019).

Hidden Markov Models (HMMs) can be used to identify a succession of subsequences in a time series (Rabiner, 1989). Rather than focusing on shifts in the mean of a process, HMMs estimate shifts in the state of a process (Whiting et al., 2004). In other words, a HMM labels the observations according to their state, which ultimately leads to a new time series with states alongside the original one with the observations. If the latter is a time series of river discharges, then the HMM will generate a new time series of climate states. In hydrology, HMMs are typically used to analyze time series exhibiting a regime-like behaviour characterized by long-term persistence (Akintug and Rasmussen, 2005; Whiting et al., 2004; Turner and Galelli, 2016).

In this article, we combine a classification obtained by an HMM with the differential split-sample testing framework. The goal is to improve the robustness of the calibration/validation of a hydrological model, which is a prerequisite to climate change impact assessment. The term "robustness" refers to the ability of the hydrological model to perform well under con-

trasted hydro-climatic conditions. This definition is coherent with the so-called robust decision-making framework that is often advocated to handle the deep uncertainty attached to climate change (Lempert et al., 2006). This is illustrated using the Senegal River Basin (SRB) as a case study. Headwaters in the SRB are still largely natural areas (Descroix et al., 2020; Faty, 2017) and the flow regime in the upper part of the basin exhibits regime-shifting behavior with departures from the inter-annual average over extended periods of time (Faye et al., 2015; Paturel et al., 2004; Dacosta et al., 2002). These characteristics makes the SRB an interesting case study to illustrate the differential split-sample testing framework with hydrologic sequences identified from an HMM.

The paper is organized as follows. Section 2 describes the methodology as well as the case study. Results are then discussed in section 3. Finally, concluding remarks are given in section 4.

## 2   Methods and material

### 2.1   Calibration and validation of a hydrological model under contrasted climates

Generally speaking, the calibration-validation of a hydrological model seeks to identify the unknown parameters of the model on one portion of historical data and then to judge the performance of the calibrated model over another portion (Roche et al., 2012). Subdividing the whole period into subsequences must be done carefully keeping in mind that the validation period must be close to the conditions to which the model will be applied operationally (Brigode et al., 2013).

Klemes (1986) proposes a hierarchical scheme for the systematic testing of hydrological models. When calibrating/validating a model under non-stationary conditions, the author recommends to follow the differential split-sample test. Depending on the nature of the change leading to non-stationary conditions, climate or land-use, the differential split sample test can take different forms. Since this paper is concerned by the robustness of hydrological models for climate change impact assessments, we focus on the differential split sample test to handle non-stationary conditions due to a changing climate. In that case, the time series of river discharges must divided into at least two stationary subsequences with contrasted climates, e.g. dry and wet, calibrate the model on one subsequence and use the other one for validation. The main idea is to make sure that the model is able to perform well under the transition required: from drier to wetter conditions or the opposite. This amounts to testing the stability of the parameters for different climate conditions (Brigode et al., 2013).

As explained in the introduction, classical rupture tests make the distinction between only two periods, therefore limiting the number of transitions that can be explored to assess the robustness of the hydrological model. This paper addresses this limitation by identifying multiple subsequences using a Hidden Markov Model (HMM), which are then use in a differential split sample testing framework.

## 2.2 Identifying stationary subsequences

Identifying multiple subsequences in a time series of river discharges comes down to detecting shifts in the flow regime, which can be done using a statistical test like the Pettitt test, or with the help of a HMM.

The non-parametric trend Pettitt test divides the streamflow record of length $T$ into two subsequences denoted $T_{pettitt.P_1}$ and $T_{pettitt.P_2}$ respectively. It involves identifying the change-point $Y$ marking the transition from one subsequence to the next. Given a random variable $q$ (e.g. annual streamflow), the Pettitt test is defined as: (Pettitt, 1979):

$$U_{t,L} = \sum_{i=1}^{t} \sum_{j=t+1}^{L} sng(q_i - q_j) \tag{1}$$

With $L$, the length of the time-serie $q$.

$$K = max(U_{t,L}) \tag{2}$$

$$p \approx 2 \times exp(\frac{-6K^2}{L^3 + L^2}) \tag{3}$$

where $K$ gives the year of the change-point if the test is significant ($p \leq 0,05$)(Pettitt, 1979).

Hidden Markov Modeling is a class of probabilistic model that can be used to label the observations (Rabiner, 1989). The motivation for adopting this type of model in hydrology is that the flow regime can be represented by a state variable that can take only a limited number of values (dry/wet for 2 states; dry/normal/wet for 3 states for examples). In other words, in parallel to the time series of historical river discharges, there exists another time series with discrete climate states. Denote

$\{q_1, q_2, ..., q_L\}$ the time series of annual flows and $\{\Phi_1, \Phi_2, ..., \Phi_L\}$ the time series of states which can only take $N$ possible values (Figure 1).

The state variable is unobserved and is accordingly referred to as a hidden variable. The hidden state $\Phi_t$ is modelled as a $N$ state Markov chain fully characterized by its transition probability matrix M with elements $M_{ij}$:

$$M_{ij} = M(\Phi_t = j | \Phi_{t-1} = i) \tag{4}$$

where $M_{ij}$ describes the transition probability to switch from the state $i$ at time $t-1$ to state $j$ at time $t$.

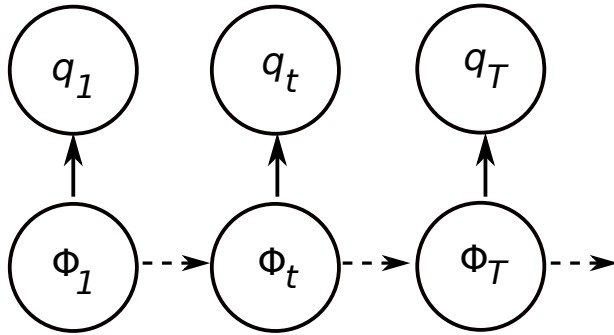

**Figure 1.** Schematic graph of Hidden Markov Modelling.

The observed variable $q_t$ is assumed to have been drawn from a probability distribution whose parameters are conditional upon the distinct state at time $t$ such that, when $\Phi_t$ is known, the distribution of $q_t$ depends only on the current state $\Phi_t$ and not on previous states or observations.

$$M(q_t|q_{t-1},...q_1,\Phi_t,...,\Phi_1) = M(q_t|\Phi_t) \tag{5}$$

A HMM is described by (1) the parameters of Gaussian distributions i.e., mean $\mu = (\mu_1, \mu_2, ..., \mu_N)$ and standard deviation $\sigma = (\sigma_1, \sigma_2, ..., \sigma_N)$ associated with N states, (2) the N×N matrix of transition probabilities M, and (3) the initial distribution of the Markov chain $\delta$. Consequently, the set of parameters to be estimated is $\theta = \{\mu, \sigma, M, \delta\}$.

Fitting a HMM to the observed sequence (here the time series of annual flows), requires evaluating the likelihood of observing that sequence, as calculated under a N-state HMM (see Appendix A for more details). In this study, we use the Expectation-Maximization (EM) algorithm, which is an iterative method for finding the maximum-likelihood estimate of the parameters of an underlying distribution when some of the data are missing. In the context of HMM, the EM algorithm is known as the Baum-Welch algorithm (Welch, 2003) and the hidden climate states are treated as missing data (Bilmes, 1998; Zucchini et al., 2017).

The EM algorithm consists of two main phases: an expectation phase called "E step", followed by a maximization phase called "M step". Given the current estimate of the HMM parameters $\theta$, the following steps are repeated until acceptable convergence is achieved: The "E step" phase of the algorithm computes the expected value of unobserved data (i.e hidden climate states) using the current estimate of the parameters and the observed data. The "M step" phase of the algorithm then provides a new estimate of the parameters by using the data from the "E step" phase as if they were actually measured data. These parameters are then used to calculate the distribution of unobserved data in the next "E step" phase of the algorithm. The resulting

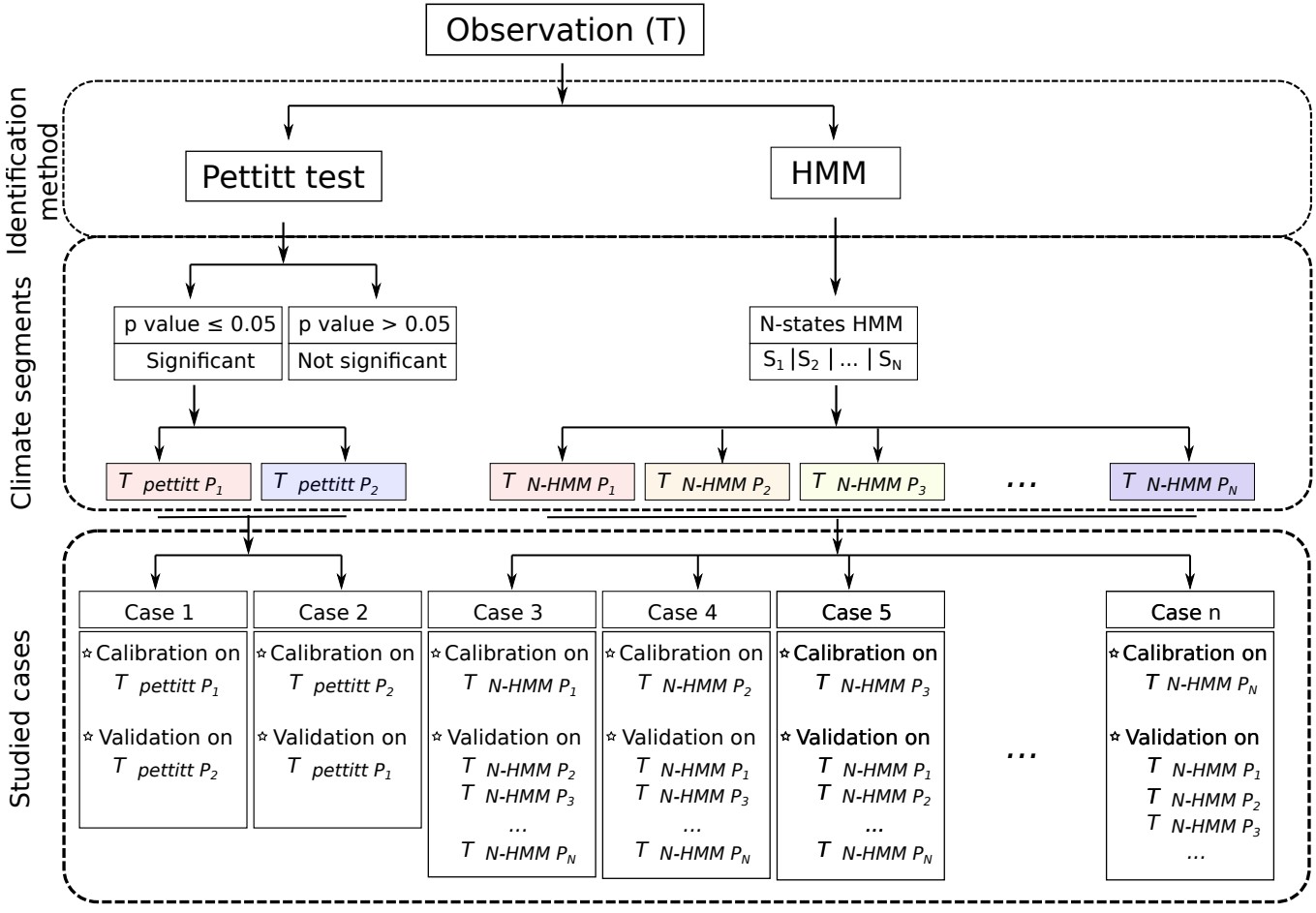

**Figure 2.** Pettitt's test and HMM identifications of flow sequences in a given period $T$. N refers to the number of states fixed by the modeller, and so the number of case available for the calibration/validation.

values of $\theta$ is then the stationary point of the likelihood of the observed data (Please refer to Appendix B for more details).

Given the observation sequence, we want to determine the sequence of hidden climate states $\{\Phi_1, \Phi_2, ..., \Phi_T\}$ that has most likely (under the fitted HMM) given rise to the time series of annual river discharges. In the literature, this is known as the decoding procedure. In this study we use the Viterbi algorithm (Viterbi, 1967) to unfold the sequence of hidden climate states

(called the Viterbi path). This, in turn, enables us to divide the whole period into numerous climate subsequences.

Figure 2 depicts the possible combinations offered by the Pettitt's test and HMM classifications. The hydrological model is calibrated on a specific subsequence, and the validation is achieved on others. Thus, the model performances (i.e. the robust-

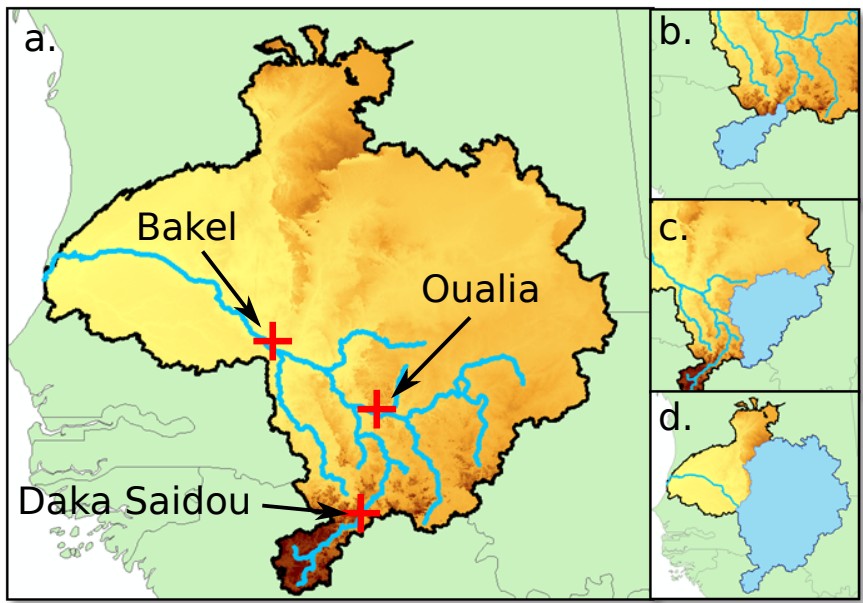

**Figure 3.** The SRB and its sub-basins boundaries. Red crosses represent sub-basin outlets (a), while sub-basin superficies are shaded in blue (b. Daka-Saidou, c. Oualia, d. Bakel).

ness) are assessed over a large panel of hydroclimatic conditions. More specifically, the robustness of the hydrological model can be assessed after examining the differences between calibration and validation scores for the different cases (transitions) that can be investigated once the subsequences are identified. If those differences remain stable, then the hydrologic model is robust vis-à-vis contrasted hydro-climatic conditions.

## 2.3 The study case: Senegal River Basin and its sub-basins

The use of HMM-derived subsequences in a differential split sample testing framework to assess the robustness of a calibrated hydrological model is illustrated with the Senegal River basin.

The Senegal River drains a basin shared by four countries in West Africa : Guinea, Mali, Mauritania, and Senegal. There are three main tributaries: (i) the Bafing River contributing to $\sim 50\%$ of the Senegal flows, (2) the Bakoye River ($\sim 15\%$), and (iii) the Faleme River (35%). Flowing northward on 500 km, the Bafing River collects precipitation on the Fouta Djallon, a high plateau considered as the water tower of West Africa. After merging with the Bakoye, the Senegal River runs north-west on 200km before the confluence with the Faleme River at Bakel, the last major tributary. After Bakel, the river meanders over 800 km through the floodplain and then discharges into the Atlantic Ocean.

| Sub-basins | River | Area (km$^2$) | Isohyets ranging (mm/y) | Outlet coordinates |
|---|---|---|---|---|
| Daka Saidou | Bafing | 15 897 | 1500-2000 | 11,96° N; 10,63° W |
| Oualia | Bakoye | 102 611 | 500-1500 | 13,61° N; 10,38° W |
| Bakel | Senegal | 393 754 | 400-2000 | 14,91° N; 12,47° W |

**Table 1.** List of the SRB sub-basins. Superficies have been calculated with the GRASS-3.4 model and 1arc sec SRTM elevation data. Indicative isohyets ranging are extracted from Faye et al. (2015).

The basin is located in the Soudano-Guinean zone, which is yearly influenced by the monsoon, a rainy season from April to October (Lahtela, 2003; Bodian, 2011). A consequence of the monsoon is a strong north/south precipitation gradient, ranging from 1900mm/y in the south to 100mm/y in the north (Bader et al., 2014; Bodian et al., 2015). In addition, precipitations present strong annual and inter-annual historical variabilities (Faye et al., 2015), with a wet episode (1950s-1970s) and a dry episode (1970s-1990s). With this historical climatic variability, as well as a strong spatial heterogeneity of its hydroclimatic components, the SRB is an interesting case study to analyze the robustness of hydrological models.

To take advantage of the hydroclimatic specificities of the SRB and its heterogeneity, we have divided the SRB into three sub-basins (Figure 3.b,c,d and Table 1). This allows us to demonstrate the potential of the proposed protocol based on an HMM classification on basins with contrasting hydrologic characteristics. Sub-basins have been delimited using the GRASS-3.4 software, and the Shuttle Radar Topography Mission (SRTM) 1arc sec elevation data set.

Generally speaking, streamflows are considered as an integrative signal of the whole basin hydro-climatic conditions, meaning that river discharges are the result of hydrological processes taking place upstream and are influenced by changes in precipitation, land use, etc. For the Senegal River Basin, most of the runoff and headwaters are located in the Fouta Djallon, a sparsely populated plateau where vegetation cover is relatively stable (Descroix et al., 2020), anthropogenic impacts on runoff seem to be negligible (Faty, 2017; Bader et al., 2014; OMVS, 2011). The areas mainly concerned with massive land-use conversions are located downstream of Bakel, a region not considered in our analysis because it marginally contributes to the river discharge. This study relies on a time series of naturalized flows at Bakel produced by Bader et al. (2014) after removing the influence of the Manantali dam. In Daka Saidou and Oualia sub-basins, however, river discharges are still natural. Consequently, we can assume that changes in the flow regime can only be attributed to shifting climate conditions.

## 2.4 The selected hydrological model

The selected hydrological model is GR2M (Mouelhi, 2003), a monthly time-step conceptual model that has already been used in the SRB with satisfactory results (Ardoin-Bardin, 2004; Ardoin-Bardin et al., 2005; Bodian et al., 2012, 2015, 2016). Moreover, since the simulated flows will be processed by a hydro-economic model of the SRB working on a monthly time step

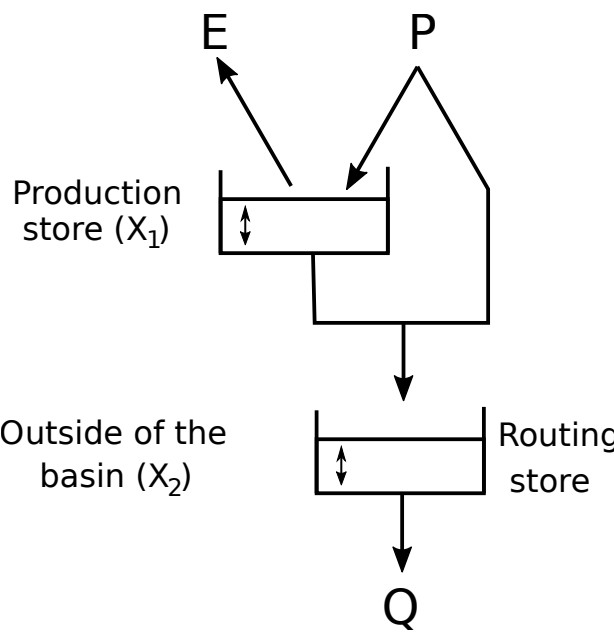

**Figure 4.** Scheme of the hydrological GR2M model.

(Tilmant et al., 2020), there was no need for a hydrological model working on a shorter time step.

GR2M has only two parameters: X1 and X2 controlling the production and the transfer functions respectively (Figure 4). We use the GR2M version included in the environment "airGR", developed by Coron et al. (2017). GR2M calibration/validation phase requires three time-series: (i) a time series of monthly precipitations (P) in the basin, (ii) a time series of monthly potential evapotranspirations (PET), and (iii) a time series of monthly river discharges (q) at the outlet.

### 2.5    Implementing the differential split sample test on HMM-derived subsequences

Many authors have pointed out that selecting the most accurate hydrological and meteorological inputs can significantly reduce the total error during the calibration/validation of a hydrological model (Paturel et al., 1995; Huard and Mailhot, 2006; Kavetski et al., 2006; Huard and Mailhot, 2008; Renard et al., 2010). Based on a comparison with meteorological observations compiled by SIEREM, and details given by Bader et al. (2014) about hydrological data, the following dataset is selected: (1) Time series of precipitations were extracted from HSM-SIEREM dataset, stretching from 1940 to 1998 (Boyer et al., 2006); (2) PET time series comes from the Climate Research Unite CRU (Harris et al., 2020), and covers a period from 1901 to 2018; (3) Monthly river discharge at sub-basin outlets are naturalized flows extracted from Bader et al. (2014) for the 1903-2012 period. Based

on the above datasets, the analysis is carried out on the period 1940-1998 (59 years), which is denoted by "the full historical record" ($T^{1940-1998}$) in the remaining of this paper.

Selecting an objective function to calibrate a conceptual hydrological model is one of the main concerns of the hydrological community (Garcia et al., 2017; Krause et al., 2005; Madsen, 2003). Here, two objective functions are selected: (1) the Nash Sutcliffe efficiency (NSE) (Nash and Sutcliffe, 1970), and the (2) Kling-Gupta Efficiency criterion (KGE) (Gupta et al., 2009). The former is a popular criterion and since it mainly focuses on high flows, it is particularly relevant for rivers where much of the annual discharge is generated during the high flow season, which is the case in the SRB. The latter allows for a multi-objective calibration that considers more components than just the errors; that is, correlation, bias and variability.

Mathematically, the NSE and KGE formulations can be written as:

$$NSE = 1 - \frac{\sum_{t=1}^{n}(q_t^{obs} - q_t^{sim})^2}{\sum_{t=1}^{n}(q_t^{obs} - \mu^{obs})^2} \tag{6}$$

$$KGE = 1 - \sqrt{(r-1)^2 + (\alpha-1)^2 + (\beta-1)^2} \tag{7}$$

with $q^{obs}$ is the observed flow at time $t$, $q^{sim}$ is the simulated flow at time $t$, $\mu^{obs}$ the mean of observed flows; $\beta$ the ratio between the mean simulated flow and the mean observed flow value $\beta = \mu^{sim}/\mu^{obs}$; $\alpha$ the ratio between the standard deviation of simulated flows and the standard deviation of observed flows $\alpha = \sigma^{sim}/\sigma^{obs}$; and $r$ is given by:

$$r = \frac{\sum_{t=1}^{n}(q_t^{obs} - \mu^{obs})(q_t^{sim} - \mu^{sim})}{\sqrt{(\sum_{t=1}^{n}(q_t^{obs} - \mu^{obs})^2) * (\sum_{t=1}^{n}(q_t^{sim} - \mu^{sim})^2)}} \tag{8}$$

For the identification of the subsequences, we have implemented the Pettitt's test, a 2-states HMM classification (a "dry" state and a "wet" state) and a 3-states HMM classification ("dry", "normal", and "wet" states). As shown on Figure 5, seven transitioning cases can be investigated within the differential split testing framework:

- If relevant, the Pettitt test offers two calibration/ validation possibilities: calibration on $T_{pettitt.dry}$ and validation on $T_{pettitt.wet}$, and vice versa.

- The 2-states HMM classification offers two possibilities too: calibration on $T_{2HMM.dry}$ and validation on $T_{2HMM.wet}$, and the opposite.

- Similarly, the 3-states HMM classification leads to three possibilities: calibration on $T_{3HMM.dry}$ and validation on $T_{3HMM.nor} + T_{3HMM.wet}$, and corollaries.

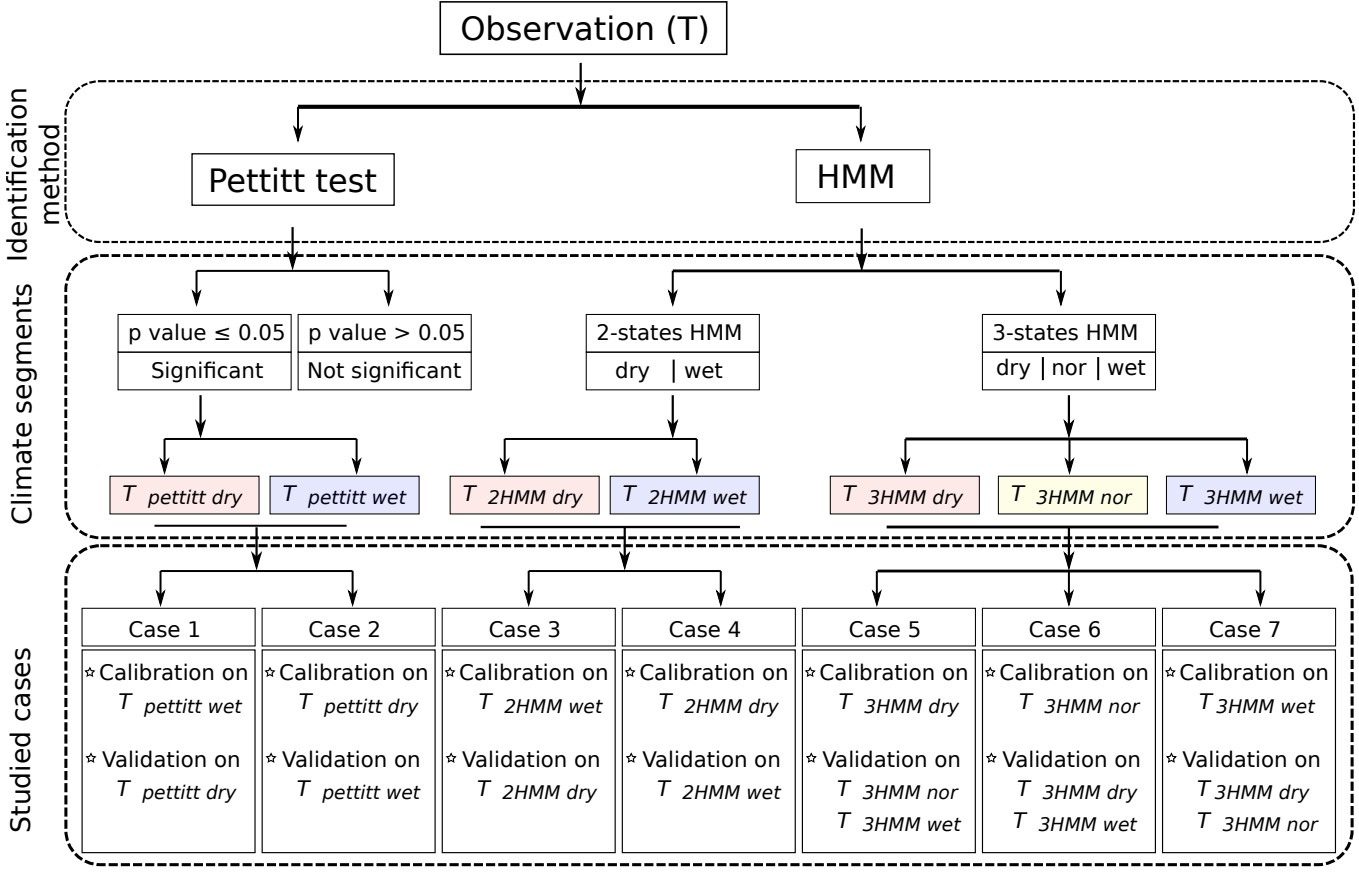

**Figure 5.** Pettitt's test and HMM identifications of flow sequences: Seven cases for the calibration/validation phase are obtained.

Pettitt test and HMM classifications have been carried out on the time series of annual flows. $T_{pettitt.wet}$ and $T_{pettitt.dry}$ are both subsequences made of contiguous years as the original time series is split in two. In that case, the temporal persistence found in the original time series is very much preserved. However, for $T_{2HMM.wet}$ and $T_{2HMM.dry}$, the situation is different since they are made of numerous, not necessarily contiguous "wet" or "dry" subsequences respectively. This is also true for $T_{3HMM.wet}$, $T_{3HMM.nor}$, and $T_{3HMM.dry}$.

Even though the KGE is based on a decomposition of the NSE, the corresponding scores cannot be directly compared. Therefore, we will discuss the results obtained with NSE and KGE separately. During all calibration phases, the first year is considered as warming-up period and not considered. Recall that the identification of change-points is done on the time series of annual flows while the hydrological model simulates monthly river discharges.

### The full historical record $T^{1940-1998}$

| | Basins | p value | Year break |
|---|---|---|---|
| Pettitt test | Daka Saidou | $1.10^{-6}$ | 1970 |
| | Oualia | $8.10^{-8}$ | 1971 |
| | Bakel | $2.10^{-6}$ | 1971 |

| | Basins | $\mu_{dry};\mu_{wet}$ | $\sigma_{dry};\sigma_{wet}$ | $\delta_{dry};\delta_{wet}$ | M |
|---|---|---|---|---|---|
| 2-states-HMM | Daka Saidou | 183.8; 300.2 | 31.3; 53.8 | 1,0 | $\begin{bmatrix} 0.97 & 0.03 \\ 0.04 & 0.96 \end{bmatrix}$ |
| | Oualia | 65.1; 178.3 | 32.1; 45.9 | 1,0 | $\begin{bmatrix} 0.968 & 0.032 \\ 0.037 & 0.963 \end{bmatrix}$ |
| | Bakel | 433.9;855.1 | 134.8;210.0 | 1,0 | $\begin{bmatrix} 0.968 & 0.032 \\ 0.037 & 0.963 \end{bmatrix}$ |

| | Basins | $\mu_{dry};\mu_{nor};\mu_{wet}$ | $\sigma_{dry};\sigma_{nor};\sigma_{wet}$ | $\delta_{dry};\delta_{nor};\delta_{wet}$ | M |
|---|---|---|---|---|---|
| 3-states-HMM | Daka Saidou | 162.5;206;300.7 | 22.8; 22.5;54.1 | 0,1,0 | $\begin{bmatrix} 0.941 & 0.059 & 0 \\ 0.063 & 0.875 & 0.063 \\ 0 & 0.04 & 0.96 \end{bmatrix}$ |
| | Oualia | 37.8;87.8;178.4 | 11.4;25.3;45.8 | 0,1,0 | $\begin{bmatrix} 0.8 & 0.2 & 0 \\ 0.125 & 0.813 & 0.063 \\ 0.037 & 0 & 0.963 \end{bmatrix}$ |
| | Bakel | 363.39; 553.32; 925.3 | 90.2; 149.0; 179.5 | 0,1,0 | $\begin{bmatrix} 0.875 & 0.125 & 0 \\ 0.188 & 0.75 & 0.062 \\ 0 & 0.056 & 0.944 \end{bmatrix}$ |

**Table 2.** Pettitt test results and Hidden Markov Model parameters (N=2 and N=3) for Daka Saidou, Oualia, and Bakel sub-basins, on the full historical record $T^{1940-1998}$.

## 3   Analysis of simulation results

First, we applied the calibration-validation protocol on the full historical record ($T^{1940-1998}$). The results (section 3.1) high-
light the relevance of an HMM classification for long time-series (59 years) with a historical contrasted climate.

Then, the protocol is implemented, independently, on two shorter periods: 1945-1971 ($T^{1945-1971}$, 27 years) and 1972-1998 ($T^{1972-1998}$, 27 years). This second test aims at illustrating the relevance of HMM classifications for shorter time series, and which do not display a clear climate trend. Results are respectively given in sections 3.2 and 3.3.

### 3.1   Subsequences identification and calibration/validation results for the full historical record $T^{1940-1998}$

The results of the division of the full historical record $T^{1940-1998}$ are displayed in Table 2 and Figure 6a. Calibration and validation values are given in Figure 6b. and in Table 3.

For the three sub-basins, the Pettitt's test is significant and shows a rupture in 1970 or 1971 (Table 2, Figure 6a. red vertical line). The 2-state HMM classification provides similar results with nearly aligned climate subsequences for all sub-basins. This

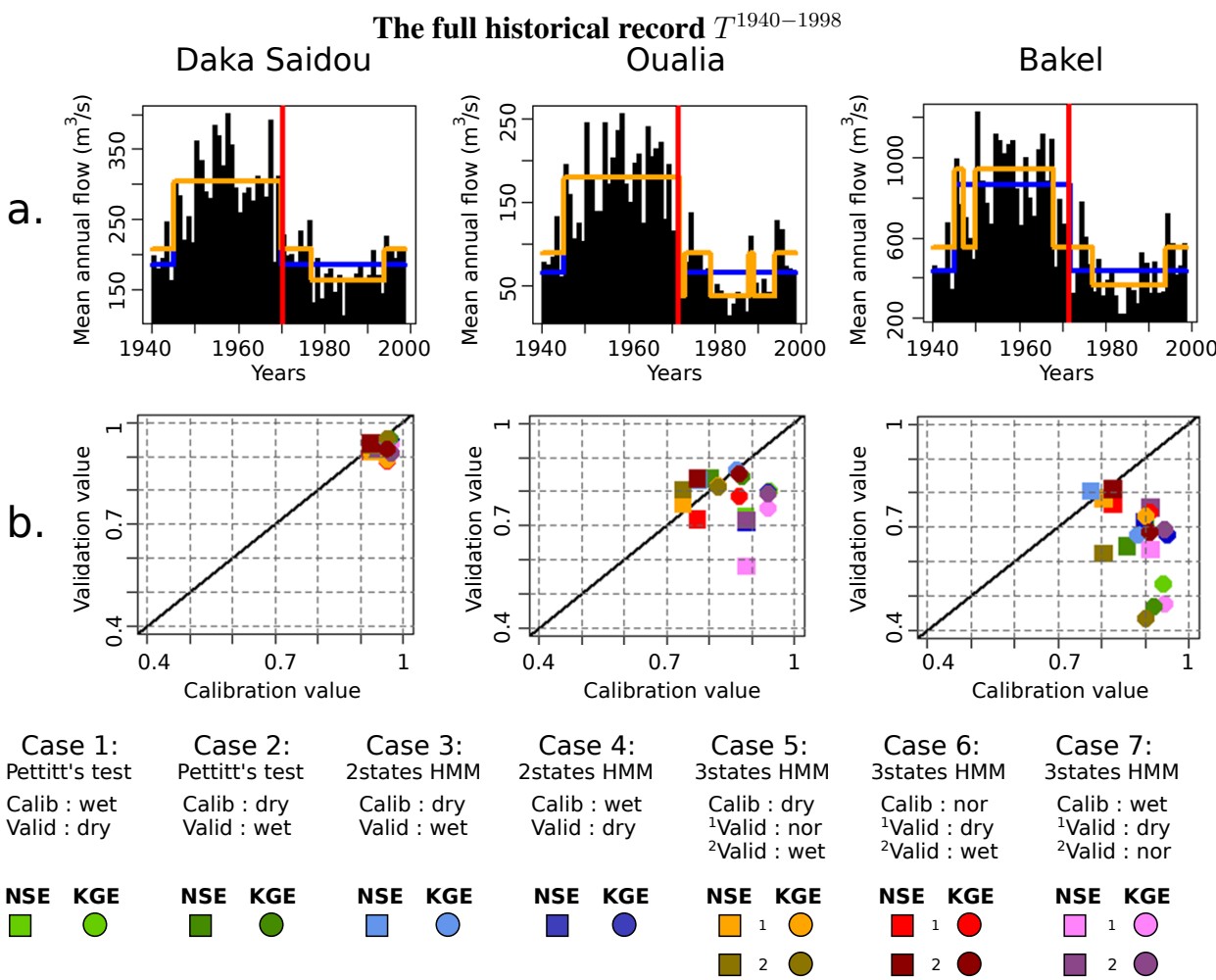

**Figure 6.** a. Classifications of $T^{1940-1998}$ according to the Pettitt test (vertical red lines), 2-states-HMM (in blue) and 3-states-HMM (in orange); b. Scatter-plot of NSE (squares) and KGE (dots) calibration/validation values. The continuous black line refer to the $Calibration_{Value} = Validation_{Value}$ line.

is also true for the 3-states HMM classification. It must also be noted that Pettitt's test-derived and HMM-derived subsequences are quite long, ranging from 15 years to 33 years.

From the examination of the transition probability matrices $M$ in Table 2, we can see that the states are clearly distinct both with the 2-states and the 3-states HMM classification. As a matter of fact, the values close to one on the diagonal indicate that when the climate is in a particular state, it will likely remain in that state in the next time period (year).

The examination of Figure 6b reveals that, for Daka Saidou (upstream), the model's scores (NSE or KGE) in calibration and validation are gathered in the top right corner, indicating that the model is able to reproduce well the subsequences of the historical record that have been used in calibration and then in validation. A statistical analysis of the subsequences shows that those sharing the same (hidden) climate state have similar statistics but that the differences across climate states are relatively small at Daka Saidou compared to the other two sub-basins located downstream (Bakel and Oualia). For example, the mean monthly streamflows of dry subsequences represent 64%, 61%, and 54% of wet subsequences' for the Pettitt test's, 2-states HMM and 3-states HMM classification respectively, which, as we will see later, are much higher ratios than those found at Bakel or Oualia. In other words, although the climate subsequences are indeed statistically distinct, they are nevertheless not that far apart, meaning that regardless of the transitions, the conditions for calibration and validation are not significantly different.

For Oualia and Bakel sub-basins, however, calibration/validation scores are more scattered; only some model versions are able to perform consistently over contrasted climates. For Oualia, calibrations on dry and normal subsequences (cases 2,3,5 and 6) provide relatively good values and similar validation scores (difference between calibration and validation scores lower than 0.1), meaning that the associated model versions could be considered as robust. Those results suggest that the "wet version" of the model struggles to simulate very dry months (especially during the dry subsequences). It seems that the "wet version" of the model does not handle well the intermittent streams which can be observed in the northern (driest) part of Oualia sub-basin during dry years. For Bakel, we can see that the calibration/validation scores obtained from the HMM-derived wet and dry subsequences (cases 3 and 4) are systematically better than those calculated for the subsequences identified by the Petittt test (cases 1 and 2). Here, calibrating on dry conditions and validating on wet does not systematically performs better than the other way around. In contrast to Oualia, Bakel sub-basin drains a portion of the Fouta Djallon with the Faleme and Bafing Rivers and is therefore less sensitive to those intermittent streams found in the north. Finally, as we move downstream, we can see that the difference between NSE and KGE scores tends to increase. As pointed out by (Gupta et al., 2009), since the NSE uses the observed mean as baseline, it can lead to overestimation of model skill for highly seasonal time series. Here, due to the north-south precipitation gradient, the seasonality of the flow regime increases once the river leaves the Fouta Djallon since it receives the contribution of more and more intermittent tributaries.

## 3.2 Subsequences identification and calibration/validation results for $T^{1945-1971}$

This section examines the period 1945-1971 ($T^{1945-1971}$), which can be considered as a wet historical episode in the SRB. The results of the division are displayed in Table 4 and Figure 7a. Calibration and validation values are given in Figure 7b. and in Table 5.

For the three sub-basins, Pettitt's tests are inconclusive, indicating that there is no clear climatic trend in the $T^{1945-1971}$ period. However, the HMM is still able to make the distinction between climate states and thus identify corresponding subsequences, which can then be exploited in a differential split-sample test. The subsequences provided by 2-state HMM and the

## The full historical record $T^{1940-1998}$

### Daka Saidou

| | subsequence(s) | Pettitt's Test | | 2-states HMM | | 3-states HMM | | |
| --- | --- | --- | --- | --- | --- | --- | --- | --- |
| | | Case 1 | Case 2 | Case 3 | Case 4 | Case 5 | Case 6 | Case 7 |
| Calibration | Dry | | 0.93/0.97(28y) | 0.92/0.96(33y) | | 0.93/0.96(17y) | | |
| | Normal | | | | | | 0.92/0.96(16y) | |
| | Wet | 0.94/0.97(30y) | | | 0.94/0.97(25y) | | | 0.94/0.97(25y) |
| Validation | Dry | 0.93/0.94(28y) | | | 0.92/0.94(33y) | | 0.91/0.89(17y) | 0.92/0.94(17y) |
| | Normal | | | | | 0.91/0.89(16y) | | 0.92/0.91(16y) |
| | Wet | | 0.93/0.95(30y) | 0.94/0.95(25y) | | 0.93/0.95(25y) | 0.94/0.92(25y) | |

### Oualia

| | subsequence(s) | Case 1 | Case 2 | Case 3 | Case 4 | Case 5 | Case 6 | Case 7 |
| --- | --- | --- | --- | --- | --- | --- | --- | --- |
| Calibration | Dry | | 0.80/0.88(27y) | 0.77/0.86(31y) | | 0.74/0.82(15y) | | |
| | Normal | | | | | | 0.77/0.87(16y) | |
| | Wet | 0.89/0.94(31y) | | | 0.89/0.94(27y) | | | 0.89/0.94(27y) |
| Validation | Dry | 0.73/0.80(27y) | | | 0.71/0.8(31y) | | 0.72/0.79(15y) | 0.58/0.75(15y) |
| | Normal | | | | | 0.76/0.82(16y) | | 0.71/0.79(16y) |
| | Wet | | 0.84/0.84(31y) | 0.83/0.86(27y) | | 0.8/0.81(27y) | 0.84/0.85(27y) | |

### Bakel

| | subsequence(s) | Case 1 | Case 2 | Case 3 | Case 4 | Case 5 | Case 6 | Case 7 |
| --- | --- | --- | --- | --- | --- | --- | --- | --- |
| Calibration | Dry | | 0.86/0.92(27y) | 0.78/0.88(31y) | | 0.8/0.9(17y) | | |
| | Normal | | | | | | 0.83/0.91(21y) | |
| | Wet | 0.9/0.94(31y) | | | 0.90/0.95(27y) | | | 0.91/0.94(20y) |
| Validation | Dry | 0.69/0.53(27y) | | | 0.71/0.68(31y) | | 0.77/0.74(17y) | 0.63/0.48(17y) |
| | Normal | | | | | 0.78/0.73(21y) | | 0.75/0.69(21y) |
| | Wet | | 0.64/0.47(31y) | 0.80/0.68(27y) | | 0.62/0.44(20y) | 0.81/0.68(20y) | |

**Table 3.** Table of NSE/KGE calibration and validation scores according to the seven cases for the full historical record $T^{1940-1998}$. The numbers of years used for the calibration or validation are given between brackets.

3-states HMM classifications are not necessary aligned for all sub-basins. As $T^{1945-1971}$ is 26-years length period, the 2-states HMM derived and the 3-states HMM derived subsequences have lengths ranging from 5 to 19 years. Various authors have discussed the minimum length required for achieve a calibration or a validation without reaching a consensus, even though a number from two to eight years could be enough depending on the "hydrological events" included in the subsequences (Razavi and Coulibaly, 2013; Juston et al., 2009; Singh and Bárdossy, 2012). In our case, the technique used to identify the subsequences (HMM classifications) seeks to provide relatively homogeneous ones. Here, we assume that five years is acceptable but we do not investigate this issue further since it is beyond the scope of our paper.

The transition probability matrices for the 2-states HMM and 3-states HMM are now diverging from an identity matrix, indicating that the temporal persistence is less pronounced than that found in the full historical records. In addition, we note that the mean annual flow of dry subsequences ($T^{1945-1971}_{2HMM.dry}$ and $T^{1945-1971}_{3HMM.dry}$) are relatively high (in comparison with $T^{1940-1998}_{2HMM.dry}$ and $T^{1940-1998}_{3HMM.dry}$).

**The wet historical episode $T^{1945-1971}$**

| Pettitt test | Basins | p value | Year break |
|---|---|---|---|
| | Daka Saidou | 0.694 | - |
| | Oualia | 0.399 | - |
| | Bakel | 0.646 | - |

| 2-states-HMM | Basins | $\mu_{dry};\mu_{wet}$ | $\sigma_{dry};\sigma_{wet}$ | $\delta_{dry};\delta_{wet}$ | M |
|---|---|---|---|---|---|
| | Daka Saidou | 273.3;366 | 25.6;43.6 | 1,0 | $\begin{bmatrix} 0.864 & 0.136 \\ 0.413 & 0.587 \end{bmatrix}$ |
| | Oualia | 124 ;201.8 | 32.1; 15 | 0,1 | $\begin{bmatrix} 0.344 & 0.656 \\ 0.287 & 0.713 \end{bmatrix}$ |
| | Bakel | 713;1073 | 65.4;125.4 | 0,1 | $\begin{bmatrix} 0.684 & 0.316 \\ 0.535 & 0.465 \end{bmatrix}$ |

| 3-states-HMM | Basins | $\mu_{dry};\mu_{nor};\mu_{wet}$ | $\sigma_{dry};\sigma_{nor};\sigma_{wet}$ | $\delta_{dry};\delta_{nor};\delta_{wet}$ | M |
|---|---|---|---|---|---|
| | Daka Saidou | 217;289.8;361.6 | 15.3;20.9;27.2 | 0,1,0 | $\begin{bmatrix} 0.589 & 0.158 & 0.252 \\ 0.141 & 0.696 & 0.163 \\ 0.154 & 0.252 & 0.594 \end{bmatrix}$ |
| | Oualia | 127.7;192.9;243.9 | 17;19.5;6.5 | 0,1,1 | $\begin{bmatrix} 0.397 & 0.281 & 0.322 \\ 0.482 & 0.435 & 0.083 \\ 0 & 0.835 & 0.165 \end{bmatrix}$ |
| | Bakel | 127.7;192.9;243.9 | 6.5;19.5;17 | 0,1,0 | $\begin{bmatrix} 0.397 & 0.281 & 0.322 \\ 0.482 & 0.435 & 0.083 \\ 0 & 0.835 & 0.165 \end{bmatrix}$ |

**Table 4.** Pettitt test results and Hidden Markov Model parameters (N=2 and N=3) for Daka Saidou, Oualia, and Bakel sub-basins, on the wet subsequence $T^{1945-1971}$.

Compared to the results associated with the full historical records and discussed in the previous section, we can see that the calibration/validation scores for the five remaining cases are more concentrated, especially for the two downstream sub-basins (Oualia and Bakel). During this wet episode, the hydrological models seems to be more robust, which is consistent with the fact that the intrinsic variability of the hydrologic conditions within that episode is smaller than the variability that characterizes the full historical records. With the north-south gradient that characterizes precipitations in the SRB, intermittent tributaries in the North under dry or normal conditions are turned into permanent rivers which are better handled by conceptual models like GR2M.

### 3.3 Subsequences identification and calibration/validation results for $T^{1972-1998}$

Here, we focus on the period 1972-1998 ($T^{1972-1998}$), which can be considered as a dry historical episode in the SRB. The results of the division and the corresponding parameters are displayed in Table 6 and Figure 8a. Calibration and validation values are given in Figure 8b. and in Table 7.

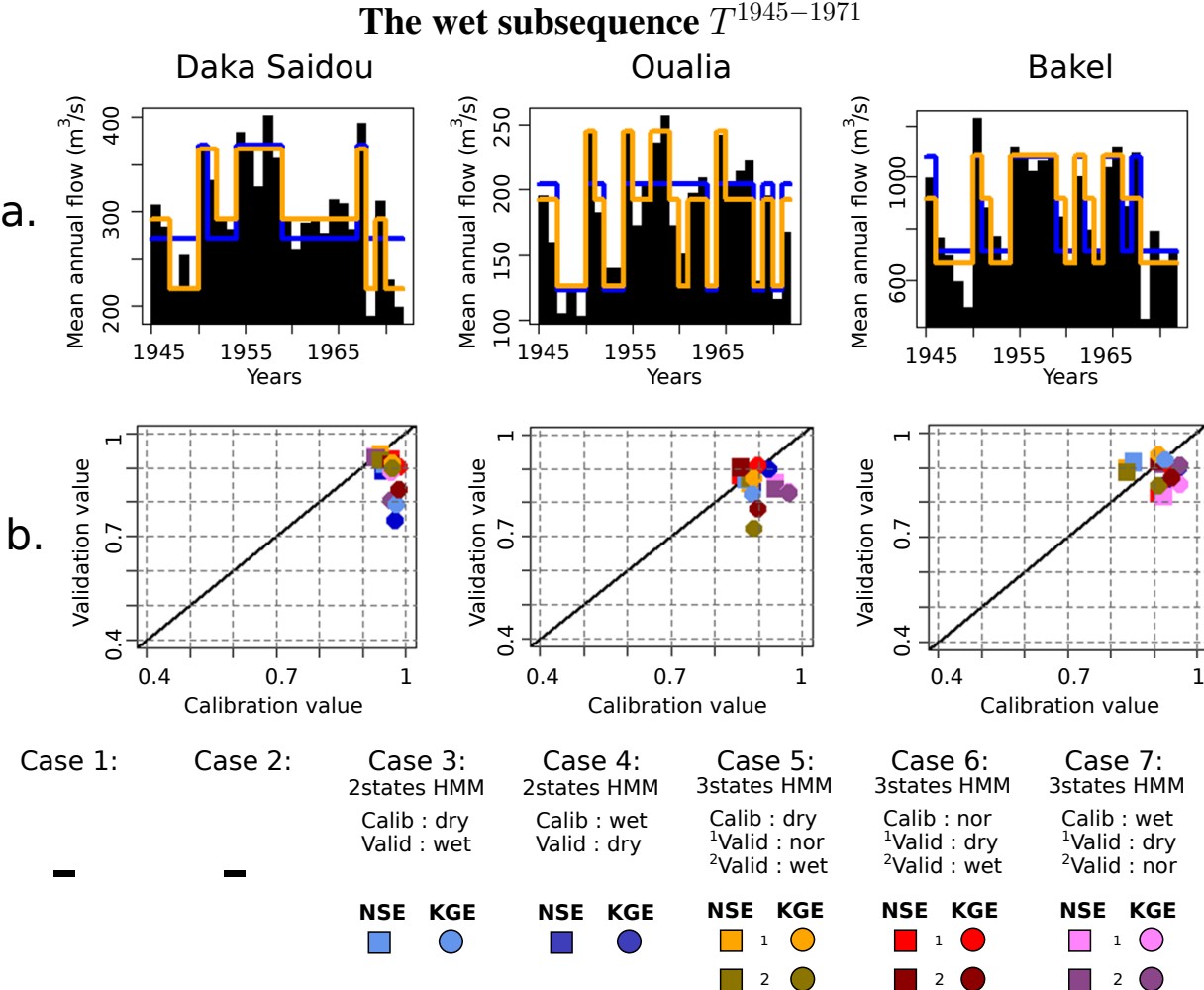

**Figure 7.** The caption is identical to the Figure 6's caption, but for the $T^{1945-1971}$ period.

Likewise in section 3.2, there is no clear monotonic climatic trend such as the Pettitt's test is inconclusive (p-values bigger than 0.05). Again, the HMM remains here a useful tool to identify subsequences, which are not necessary aligned for all sub-basins.

For Daka Saidou, all calibration and validation scores are higher than 0.9 and the differences are small (below 0.1). For Oualia sub-basin, the calibration and validation of the hydrological model face the typical challenges associated with the high spatial and temporal variability that characterizes the formation and propagation of river flows in dryland regions. The dry episode, centered around the 80ies, was triggered by a sustained reduction in precipitations (Faye et al., 2015), which was even more pronounced in the North where some tributaries became intermittent rivers (Bader et al., 2014). Conceptual hydrological models like GR2M are indeed not well equipped to deal with with sudden and widespread transitions from wet to dry

# The wet subsequence $T^{1945-1971}$

## Daka Saidou

| | subsequences(s) | Pettitt's Test | | 2-states HMM | | 3-states HMM | | |
| --- | --- | --- | --- | --- | --- | --- | --- | --- |
| | | Case 1 | Case 2 | Case 3 | Case 4 | Case 5 | Case 6 | Case 7 |
| Calibration | Dry | | */* | 0.95/0.98(19y) | | 0.94/0.97(6y) | | |
| | Normal | | | | | | 0.97/0.98(12y) | |
| | Wet | */* | | | 0.95/0.97(7y) | | | 0.93/0.96(8y) |
| Validation | Dry | */* | | | 0.89/0.75(19y) | | 0.92/0.9(6y) | 0.93/0.88(6y) |
| | Normal | | | | | 0.94/0.91(12y) | | 0.93/0.81(12y) |
| | Wet | | */* | 0.91/0.79(7y) | | 0.92/0.9(8y) | 0.91/0.83(8y) | |

## Oualia

| | subsequences(s) | Case 1 | Case 2 | Case 3 | Case 4 | Case 5 | Case 6 | Case 7 |
| --- | --- | --- | --- | --- | --- | --- | --- | --- |
| Calibration | Dry | | */* | 0.87/0.88(8y) | | 0.88/0.89(9y) | | |
| | Normal | | | | | | 0.86/0.9(12y) | |
| | Wet | */* | | | 0.89/0.92(18y) | | | 0.94/0.97(5y) |
| Validation | Dry | */* | | | 0.86/0.9(8y) | | 0.89/0.91(9y) | 0.85/0.83(9y) |
| | Normal | | | | | 0.85/0.87(12y) | | 0.84/0.83(12y) |
| | Wet | | */* | 0.87/0.82(18y) | | 0.87/0.72(5y) | 0.9/0.78(5y) | |

## Bakel

| | subsequences(s) | Case 1 | Case 2 | Case 3 | Case 4 | Case 5 | Case 6 | Case 7 |
| --- | --- | --- | --- | --- | --- | --- | --- | --- |
| Calibration | Dry | | */* | 0.85/0.92(16y) | | 0.84/0.91(12y) | | |
| | Normal | | | | | | 0.91/0.94(5y) | |
| | Wet | */* | | | 0.92/0.96(10y) | | | 0.92/0.96(9y) |
| Validation | Dry | */* | | | 0.85/0.9(16y) | | 0.82/0.9(12y) | 0.81/0.85(12y) |
| | Normal | | | | | 0.9/0.94(5y) | | 0.91/0.91(5y) |
| | Wet | | */* | 0.91/0.92(10y) | | 0.88/0.85(9y) | 0.92/0.87(9y) | |

**Table 5.** Table of NSE/KGE calibration and validation scores according to the seven cases for the wet subsequence $T^{1945-1971}$. As the Pettitt's test is not conclusive here, no calibration/validation scores are given (symbols $*/*$).

conditions (Gutierrez-Jurado et al., 2021). For Bakel, no clear pattern emerges: some "dry-versions" have as bad (or good) performances as some "wet versions". The poorest scores are nevertheless obtained when the calibration and validation are carried out on homogeneous subsequences associated to extreme climate states (dry - wet, wet-dry), e.g. cases 5-2 and 7-1.

## 4  Conclusions

This article proposes an HMM-based classification to deal with complex climate sequences and shows how the resulting classification can be used in a differential split sample test to assess the robustness of a hydrological model. A modeling experiment is carried out in the Senegal River basin using the GR2M model and historical flows from 1940-1998. Then, two other periods have been investigated, a wet episode (1945-1971) and a dry one (1972-1998).

The main concluding remarks are:

## The dry historical episode $T^{1972-1998}$

| | Basins | p value | Year break | | |
|---|---|---|---|---|---|
| Pettitt test | Daka Saidou | 0.277 | - | | |
| | Oualia | 0.399 | - | | |
| | Bakel | 0.474 | - | | |

| | Basins | $\mu_{dry};\mu_{wet}$ | $\sigma_{dry};\sigma_{wet}$ | $\delta_{dry};\delta_{wet}$ | M |
|---|---|---|---|---|---|
| 2-states-HMM | Daka Saidou | 162.4;210.1 | 18.1;22.8 | 0,1 | $\begin{bmatrix} 0.933 & 0.067 \\ 0.128 & 0.872 \end{bmatrix}$ |
| | Oualia | 37.8;88.4 | 11.4; 25.8 | 1,0 | $\begin{bmatrix} 0.781 & 0.219 \\ 0.185 & 0.814 \end{bmatrix}$ |
| | Bakel | 356.9;668.2 | 111.3;81.3 | 1,0 | $\begin{bmatrix} 0.903 & 0.097 \\ 0.487 & 0.513 \end{bmatrix}$ |

| | Basins | $\mu_{dry};\mu_{nor};\mu_{wet}$ | $\sigma_{dry};\sigma_{nor};\sigma_{wet}$ | $\delta_{dry};\delta_{nor};\delta_{wet}$ | M |
|---|---|---|---|---|---|
| 3-states-HMM | Daka Saidou | 132.7; 173.9 ;210.8 | 12.3; 12.8;18 | 0,0,1 | $\begin{bmatrix} 0 & 0.757 & 0.243 \\ 0.307 & 0.693 & 0 \\ 0.129 & 0 & 0.871 \end{bmatrix}$ |
| | Oualia | 37.9; 69.2; 105 | 3.1;11.3;21.4 | 1,0,0 | $\begin{bmatrix} 0.799 & 0 & 0.201 \\ 0.263 & 0.736 & 0 \\ 0.141 & 0.276 & 0.583 \end{bmatrix}$ |
| | Bakel | 315.6; 421.3; 686.1 | 44.2; 71.1; 93.5 | 1,0,0 | $\begin{bmatrix} 0.482 & 0.518 & 0 \\ 0.756 & 0 & 0.244 \\ 0 & 0.5 & 0.5 \end{bmatrix}$ |

**Table 6.** Pettitt test results and Hidden Markov Model parameters (N=2 and N=3) for Daka Saidou, Oualia, and Bakel sub-basins, on the dry subsequence $T^{1972-1998}$.

– When records display a single point change, a classical rupture trend (as Pettitt test) remains an adequate tool to divide the records into two climate subsequences.

– If the records contain multiple change points, an HMM classification can divide the series into several climate subsequences without the need for additional data.

– Regardless of the division method used, the range of climate conditions over which the hydrological model can perform depends on the intrinsic variability of the series. Compared to the Pettitt test, however, the HMM classification allows for a finer labelling of the years, therefore better exploiting the intrinsic variability in the series to enrich a differential split sample test.

– We encourage the modellers to explore as many cases as possible to calibrate/validate a hydrological model according to the differential split-sample test. The parameter's stability over contrasted hydro-climatic conditions seems to depend on the studied period, on the objective functions, on the subsequences identification techniques, and the basin.

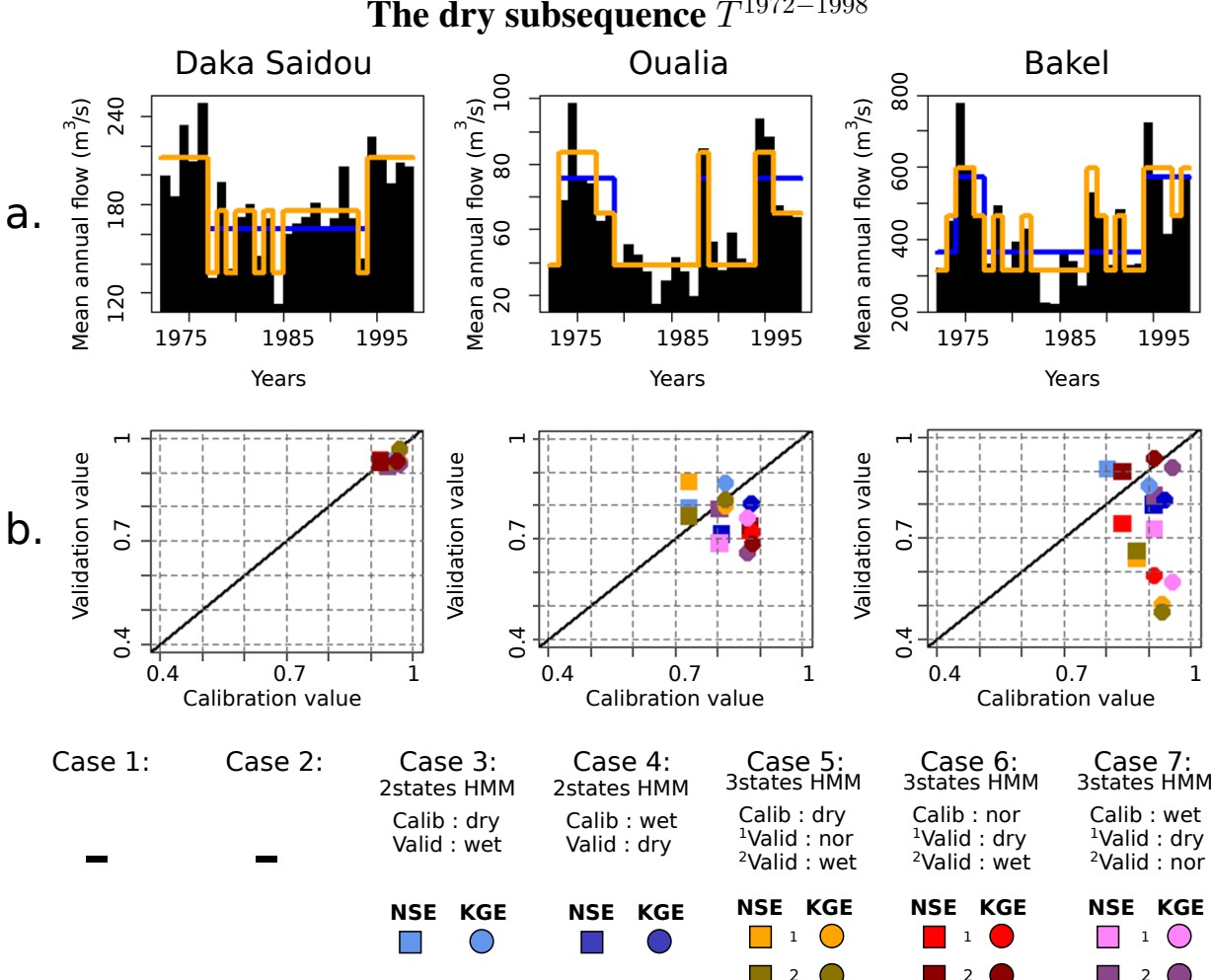

**Figure 8.** The caption is identical to the Figure 6's caption, but for the $T^{1972-1998}$ period.

## Appendix A: Likelihood of Hidden Markov Models

We suppose there is an observation sequence $Q = \{q_1, q_2, ..., q_T\}$ and the associated (unobserved) state variables $\Omega = \{\Phi_1, \Phi_2, ..., \Phi_T\}$ generated by such a model. Given the set of HMM parameters $\theta = \{\mu, \sigma, M, \delta\}$, the joint density of complete data set $Z = (Q, \Omega)$ can be expressed as:

$$p(Z|\theta) = p(Q, \Omega|\theta) = p(Q|\Omega, \theta)p(\Omega|\theta) \tag{A1}$$

# The dry subsequence $T^{1972-1998}$

## Daka Saidou

| | | Pettitt's Test | | 2-states HMM | | 3-states HMM | | |
| | subsequence(s) | Case 1 | Case 2 | Case 3 | Case 4 | Case 5 | Case 6 | Case 7 |
|---|---|---|---|---|---|---|---|---|
| Calibration | Dry | | */* | 0.93/0.96(17y) | | 0.94/0.97(5y) | | |
| | Normal | | | | | | 0.92/0.96(12y) | |
| | Wet | */* | | | 0.94/0.97(9y) | | | 0.94/0.97(9y) |
| Validation | Dry | */* | | | 0.92/0.93(17y) | | 0.94/0.94(5y) | 0.93/0.97(5y) |
| | Normal | | | | | 0.92/0.93(12y) | | 0.93/0.81(12y) |
| | Wet | | */* | 0.93/0.94(9y) | | 0.93/0.97(9y) | 0.93/0.93(9y) | |

## Oualia

| | | Pettitt's Test | | 2-states HMM | | 3-states HMM | | |
| | subsequence(s) | Case 1 | Case 2 | Case 3 | Case 4 | Case 5 | Case 6 | Case 7 |
|---|---|---|---|---|---|---|---|---|
| Calibration | Dry | | */* | 0.73/0.82(14y) | | 0.73/0.82(14y) | | |
| | Normal | | | | | | 0.88/0.88(5y) | |
| | Wet | */* | | | 0.81/0.88(12y) | | | 0.8/0.87(7y) |
| Validation | Dry | */* | | | 0.71/0.8(14y) | | 0.72/0.72(14y) | 0.69/0.76(14y) |
| | Normal | | | | | 0.87/0.8(5y) | | 0.79/0.66(5y) |
| | Wet | | */* | 0.79/0.87(12y) | | 0.77/0.82(7y) | 0.72/0.68(7y) | |

## Bakel

| | | Pettitt's Test | | 2-states HMM | | 3-states HMM | | |
| | subsequence(s) | Case 1 | Case 2 | Case 3 | Case 4 | Case 5 | Case 6 | Case 7 |
|---|---|---|---|---|---|---|---|---|
| Calibration | Dry | | */* | 0.73/0.82(18y) | | 0.73/0.82(12y) | | |
| | Normal | | | | | | 0.88/0.88(7y) | |
| | Wet | */* | | | 0.81/0.88(8y) | | | 0.8/0.87(7y) |
| Validation | Dry | */* | | | 0.71/0.8(18y) | | 0.72/0.72(12y) | 0.69/0.76(12y) |
| | Normal | | | | | 0.87/0.8(7y) | | 0.79/0.66(7y) |
| | Wet | | */* | 0.79/0.87(8y) | | 0.77/0.82(7y) | 0.74/0.68(7y) | |

**Table 7.** Table of NSE/KGE calibration and validation scores according to the seven cases for the dry subsequence $T^{1972-1998}$. As the Pettitt's test is not conclusive here, no calibration/validation scores are given (symbols $*/*$).

Assuming the data belonging to each hidden state are characterized by a specific Gaussian probability distribution, the two terms on the right-hand side are:

$$p(Q|\Omega,\theta) = \prod_{t=1}^{T} p(q_t|\mu_{\Phi_t}, \sigma_{\Phi_t}) \tag{A2}$$

$$p(\Omega|\theta) = \delta \prod_{t=1}^{T-1} p((\mu_{\Phi_{t+1}}|\sigma_{\Phi_t})|M) \tag{A3}$$

The complete data likelihood function $\zeta(\theta|Z)$ can be calculated as:

$$\zeta(\theta|Z) = \zeta(\theta|Q,\Omega) = p(Q,\Omega|\theta) \tag{A4}$$

For a HMM which has the initial distribution $\delta$ and transition probability matrix M for the Markov chain, let us define the probability mass function of Q if the Markov chain is in state i at time t as:

$$p_i(q) = p(Q = q | \Omega = i) \tag{A5}$$

With $i = 1, 2, ... N$

The general form of likelihood function is then given by (Zucchini et al., 2017):

$$\zeta = \delta \Gamma(q_1) M \Gamma(q_2) ... M \Gamma(q_T) 1' \tag{A6}$$

where $\Gamma(q)$ is defined as the diagonal matrix with i the diagonal element $p_i(q)$ and $1'$ is N dimensional vector of 1.

## Appendix B:  HMM Likelihood maximization with EM algorithm

In order to set out the likelihood computation in the form of Baum-Welch algorithm (Welch, 2003), which involves the forward $\alpha(t)$ and backward $\beta(t)$ probabilities, we define $\alpha(t)$ and $\beta(t)$ as:

$$\alpha(t) = \delta \Gamma(q_1) M \Gamma(q_2) ... M \Gamma(q_t) = \delta \Gamma(q_1) \prod_{n=2}^{t} M \Gamma(q_n) \tag{B1}$$

and

$$\beta(t) = \delta \Gamma(q_{t+1}) M \Gamma(q_{t+2}) ... M \Gamma(q_t) 1' = \left( \prod_{n=t+1}^{T} M \Gamma(q_n) \right) 1' \tag{B2}$$

respectively. More specifically, $\alpha_i(t)$ is the probability of observing the partial sequence $q_1, q_2, ..., q_t$ and ending up in state i at time t, and $\beta_i(t)$ is the probability of observing the remaining sequence. Numerical calculation of $\alpha_i(t)$ and $\beta_i(t)$ is not trivial (Akintug and Rasmussen, 2005). Here we use the method suggested by Durbin et al. (1998) for scaling forward and backward probabilities to overcome this problem. Now let us define $u_j(t)$ and $v_{jk}(t)$ as (Zucchini et al., 2017):

$$u_j(t) = p(\Phi_t = j | Q, \theta) = \frac{\alpha_j(t) \beta_j(t)}{\zeta} \tag{B3}$$

$v_{jk}(t) = p(\Phi_{t-1} = j, \Phi_t = k | Q) = \alpha_j(t-1) M_{jk} p_k(q_t) \beta_k(t) / \zeta \tag{B4}$

Where $M_{jk}$ is the probability of transition from hidden climate state j to climate state k, and $\zeta$ is the likelihood function. With EM algorithm, we aim to maximize the log-likelihood of the parameters of interest $\theta$, based on complete data (i.e. both

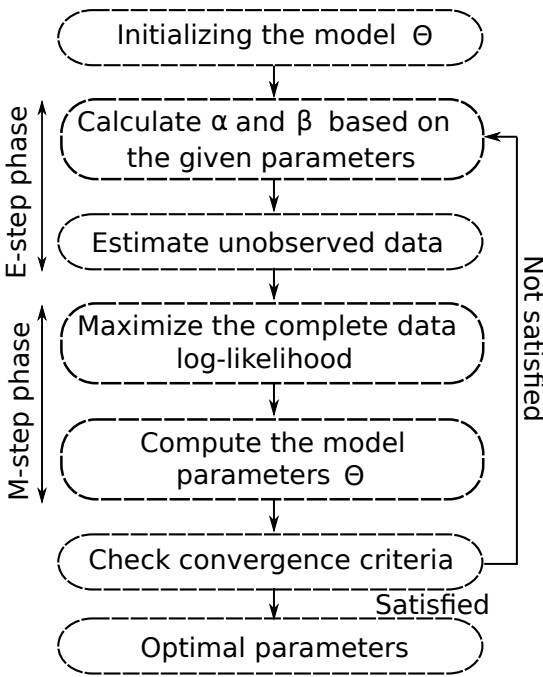

**Figure B1.** Expectation Maximization algorithm for a HMM parameter estimation.

the observed data and the hidden climate states). Now let us represent the sequence of climate states (missing data) by the Markov chain by the zero-one random variables. The complete data log-likelihood can be formulated as:

$$390 \quad \log(\zeta(\theta|Z)) = \sum_{j=1}^{N} u_j(1)\log(\delta_j) + \sum_{j=1}^{N}\sum_{k=1}^{N}(\sum_{t=2}^{T} v_{jk}(t))\log(M_{jk}) + \sum_{j=1}^{N}\sum_{t=1}^{T} u_j(t)\log(p_j(q_t)) \tag{B5}$$

where $u_j(t) = 1$ if and only if $\Phi_t = j(t = 1, 2, ..., T)$, and transition probability $v_{jk}(t) = 1$ if and only if $\Phi_{t-1} = j$ and $\Phi_t = k(t = 2, 3, ..., T)$, N is the number of hidden climate states,$\delta_j$ is the initial transition of Markov chain, and $p_j(.)$ is the probability mass function if the Markov chain is in state $j$ at time $t$. Maximization of the complete data log-likelihood function is performed with the EM algorithm through an iterative process presented in Figure B1.

*Author contributions.* A. Tilmant suggested the integration of HMM classifications into a calibration/validation protocol. V. Espanmanesh ran the HMM onto flows to provide climate subsequences. E. Guilpart carried out all the calibrations and validations, and write this paper (excepted the a portion of section 2.2 and Appendix A and B which were written by V. Espanmanesh). A. Tilmant supervised the writing. F. Anctil brought his points of view and proofred the paper.

*Acknowledgements.* The work was supported by a project from the Food and Agriculture Organization of the United Nations (FAO) entitled SAGA "Sécurité Alimentaire: une Agriculture adaptée".

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
