# Peer review of "Combining split-sample testing and Hidden Markov Modeling to assess the robustness of hydrological models"

_Hydrology and Earth System Sciences, 2020_

## Author Comment (AC1) · 8 Jan 2021

Dear all,

After one more careful reading, I denoted a (very) small mistake in table-4. In the yellow box for 3-state HMM, there is written "T 3HMM dry", while it should be "T 3HMM nor". As recommended by our editorial support, we wish to inform all of you about this mistake.

We apologize for the inconvenience. Please, accept all our warmest regards.

Etienne Guilpart

---

## Referee Comment (RC1) · Anonymous Referee #1 · 25 Jan 2021

- Comments : In this study, it is interesting to apply the HMM to find hidden states in a method for identifying hydro-climatic condition of data used to calibrate/validate hydrological model. It is expected that the method of identifying annual hydro-climatic states by generating hidden state sequences through HMM will be more systematic and useful. However, to improve the completeness of this paper, several supplements are needed as follows:

(1) Apart from the comparison between the Petitt's test and the HMM, it is necessary to present a comparative analysis of whether the climate classification sequence identified by the HMM reflects temporal variations in other meteorological data or land

use. For example, it would be possible to present any changes in land use or to state whether the temporal behavior of the dry index from annual rainfall and reference evapotraspiration over the same period is similar to the sequence of climatic state identified by the HMM.

(2) The three sub-basin are all located within one same basin. So their flow data show similar temporal behavior with different scale. This makes it difficult to generalize the results of this study. Moreover they show same dramatic changes in climatic conditions. This is rather thought to make it difficult to show the advantages of the proposed method in this study. In the abstract section, the authors mentioned that the results show that when the time series of river discharges does not exhibit a clear climate trend, or when it has multiple change points, classical rupture tests are useless and HMM classification is a viable alternative as long as the climate sub-sequences are long enough. However, the results in section 5 do not adequately explain this. The results show that Pettitt's test is still on of the appropriate tools. Perhaps an addition of another time series (basin) should be considered that clearly illustrates the difference between methods.

(3) The ultimate goal in hydrological modeling would be to obtain a better fit. The HMM's theoretical advantages of more granular and continuous identification is understood, but the results do not seem to support it. The authors noted in Section 5.2 that the HMM could lead to better model performances than the Pettitt's test, but it is difficult to accept the argument that the HMM is a better way with the values provided in Table 3. I think each method has similar NSE (KGE) values. A clearer rationale or explanation is needed for this part.

(4) The sentence for the length of data mentioned in section 6 is not the result of this paper. It was only cited from other paper and no substantive analysis was performed to support this conclusion. A minimal analysis needs to be performed to apply the claims of existing studies to the method proposed in this study.

**HESSD**

(5) Please check some typos. ex. Check the isohyets range on the Bakel basin (Table 1). ex. Check T3HMMnor in Climate segments (Figure 4).

───────────────────────

---

## Author Comment (AC2) · 13 Feb 2021

We would like to thank you for carefully reading our manuscript and providing us with valuable comments. We please to invite you to read the first paragraph underlining the context in which the paper has been achieved. Then, you will find our point-by-point response to your comments.

Paper context: The work was supported by a project from the Food and Agriculture Organization of the United Nations (FAO) entitled SAGA "Sécurité Alimentaire: une Agriculture adaptée". In this project; we focused on the Senegal River basin. A complete modeling chain has been set involving climate projections, hydrological projections, and hydro-economic modeling. The goal was to assess the relevance of existing and planned infrastructures (such as dams and irrigated areas) in the light of climate changes. Consequently, we developed this calibration/validation protocol for the hydrological model in order to produce hydrological projections. That is why the studied area was limited to the Senegal River basin.

Responses to your comments: (i) River discharges could be seen as integrative signals of all hydrological processes in the basin, and applying Pettitt's test or HMM classification on it leads to detect either climate segments or land uses switching. We completely agree that a comparison between Pettitt's test/HMM classification on river discharge and a climate variable (such as precipitation) would be relevant to distinguish climate sequences from land-use changes. In that sense, some "hydrological anomalies" (as mentioned in Savenije 2009) could be highlighted by a more formal process. However, this interesting point is not a part of the calibration/validation protocol stricto sensu, and we could consider adding a new section in the supplementary material to tackle it depending on the reviewer's comments.

(ii a) It is true that additional basins in another part of the world would permit better generalize the results of this study. As mentioned in the "Paper context" paragraph, here we limited our study to the Senegal River basin because this paper is a part of the large project SAGA. Also, even if we agree adding other illustration cases would bring more clearness to this paper, it is not strictly speaking our scientific mandate. (ii b) The section only focuses on the 1940-1998 period, which displays clear trends for each sub-basin. To underline the relevance of HMM classification in a not-clear trend situation, we added in Supplementary Materials two sections focusing on shorter periods (26 years long periods, 1945-1971 and 1972-1998). These results allow us to discuss the length of the period.

(iii) Please accept our apologies if, after careful reading, the main focus of this article has passed out of scope. The goal is to improve the robustness of the calibration/validation of a hydrological model by evaluating its performance under a large

panel of climate conditions. Strictly speaking, that does not necessarily imply obtaining a better fit, but reaching NSE or KGE values for dry/wet or dry/normal/wet sub-periods, and thus getting an idea of the performance of the model under these conditions. So, we thank you for underlining this point, and we will clarify it according to the reviewer's comments.

(iv) Indeed, a full analysis of the minimum length for a period is not carried out. However, you could find in Supplementary Materials two sections dealing with shorter periods (as mentioned above). Please, let us know if these two sections could be considered as a minimal analysis to support the statement in conclusion, or a deeper analysis would be required.

(v) We particularly appreciate your careful reading, and we will proofread all the paper to check for the mentioned mistakes and for any mistakes.

Reference : Savenije, H. H. G.: HESS Opinions "The art of hydrology"*, Hydrol. Earth Syst. Sci., 13, 157–161, https://doi.org/10.5194/hess-13-157-2009, 2009.

―――――――――――――

---

## Referee Comment (RC2) · Anonymous Referee #2 · 17 Feb 2021

The study applies the differential split-sample test to evaluate the robustness of hydrological model parameters using the HMM to identify sub-sequences of different hydroclimatic conditions. The approach is tested over three sub-basins of the Senegal River employing the GR2M model; the 2-states and 3-states HMM classifications are compared with the non parametric Pettitt's test which allows to detect a single change point. Authors state that results show that HMM can be a viable classification option for long time series exhibiting multiple change points.

There are several points that deserve further insights and that are not sufficiently explored in the study. 1) The title of the paper hints at a protocol of calibration/validation

of hydrological models that in my opinion is still substantially based on the differential split sample test proposed by Klemes (1986). The study rather focuses on the comparison of different approaches to identify sub-periods characterized by different climatic conditions. The 3steps protocol outlined at sec. 4 is not novel and should at least include calibration and validation steps. I suggest rephrasing the title.

2) I agree with remark in RC1 about the necessity of a more detailed analysis to support the climate classification sequence identified by the HMM in terms of changes in meteorological data or land use.

3) Results in Table 3 are too shortly discussed in Section 5. Although HMM allows for a finer labelling of each year, NSE and KGE coefficients in Table 3 do not show higher performance in fitting observed flows. Classification based on Pettitt's test provides comparable or better results, particularly for the Daka Saidou and Oualia basins. An interesting insight could be given on which is the most convenient case to apply to reach better performance (Case 1 vs Case 2; Case 3 vs Case 4;….). The case study and the results do not allow a generalization of the approach and probably it is not the ideal test case to show the advantages of the proposed method mostly because there is a marked rupture in streamflow observations.

4) The content of the final paragraph at sect. 5.1 is not clear and differences (if any) with the traditional differential split sample test should be better outlined. Moreover, conclusions not always refer to by presented analysis and results showed in the study and do not seem to support statement in the abstract.

There are some other minor comments (see below) that I recommend to consider.

Table 1: Check isohyets ranging for the Bakel sub-basin.

Table 2: can be improved by associating parameters to dry, wet and nor period.

Figure 5b: I suggest to use the same axis limits for both validation and calibration performances.

Eq. (3): Check equation symbol (*)

Eq. (4, 5, 6): several symbols are not introduced or explained in text.

Line 205: should be T3HMMnor

Recent examples of differential split-sample validation tests that can be included as references have been reported in:

D.F. Motavitaab, R. Chowab, A. Guthkea, W. Nowaka. (2019). The comprehensive differential split-sample test: A stress-test for hydrological model robustness under climate variability, Journal of Hydrology, Volume 573, Pages 501-515

H. Dakhlaouiab, D. Ruellandc, Y.Tramblayd. (2019) A bootstrap-based differential split-sample test to assess the transferability of conceptual rainfall-runoff models under past and future climate variability, Journal of Hydrology, Volume 575, Pages 470-486

---

## Author Comment (AC4) · 17 Mar 2021

We highly appreciate your time and effort in conducting a thorough review of our manuscript. Please find below our responses (your comments are in bold, our responses in normal font, and proposed changes to the manuscript in italic).

The study applies the differential split-sample test to evaluate the robustness of hydrological model parameters using the HMM to identify sub-sequences of different hydroclimatic conditions. The approach is tested over three sub-basins

of the Senegal River employing the GR2M model; the 2-states and 3-states HMM classifications are compared with the non parametric Pettitt's test which allows to detect a single change point. Authors state that results show that HMM can be a viable classification option for long time series exhibiting multiple change points.

There are several points that deserve further insights and that are not sufficiently explored in the study.

(1) The title of the paper hints at a protocol of calibration/validation of hydrological models that in my opinion is still substantially based on the differential split sample test proposed by Klemes (1986). The study rather focuses on the comparison of different approaches to identify sub-periods characterized by different climatic conditions. The 3steps protocol outlined at sec. 4 is not novel and should at least include calibration and validation steps. I suggest rephrasing the title.

We propose the following title :

Combining split-sample testing and Hidden Markov Modeling to assess the robustness of hydrological models.

(2) I agree with remark in RC1 about the necessity of a more detailed analysis to support the climate classification sequence identified by the HMM in terms of changes in meteorological data or land use.
We agree that river discharges are the result of hydrological processes taking place upstream and are influenced by changes in precipitation, land use, etc. Anthropogenic changes may indeed alter the flow regime and hence influence an HMM-derived classification which would no longer rely solely on natural factors. However, that anthropogenic influence can be removed so that the analysis is done on naturalized flows, something fairly standard in time series analysis but also in process-based hydrological modelling.

The overall goal of this paper is to highlight the relevance of combining HMMs classifications with the differential split-sample test to assess the robustness of a hydrological model to be used in climate change studies, i.e. to generate hydrologic projections from contrasted climate ones. So, the emphasis is on changes in physical processes rather than human-induced ones. The Senegal River basin is only used as a case study to illustrate the proposed method to test the robustness of a hydrological model. For that river basin, most of the runoff and headwaters of two of the three sub-basins (Daka Saidou, Oualia) are located in the Fouta Djallon, a sparsely populated plateau where vegetation cover is relatively stable (Descroix et al, 2020), anthropogenic impacts on runoff seem to be negligible (Faty, 2017) or not even mentioned in the updated Senegal River monography (Bader et al, 2014) and in the River Basin Master Plan (OMVS, 2011). The areas mainly concerned with massive land-use conversions are located downstream of Bakel, a region not considered in our analysis. For the third sub-basin, river discharges at the outlet were naturalized by Bader et al. (2014) after removing the influence of the Manantali dam on the flow regime.

In this study, the term "robustness" refers to the ability of the hydrological model to perform well under contrasted hydro-climatic conditions. This definition is coherent with the so-called robust decision-making framework that is often advocated to handle the deep uncertainty attached to climate change (Lempert et al., 2006). In other
words, we do not seek the "best" model (with the best fit) but a model that performs reasonably well under different conditions.

Also, we propose replacing the second to last paragraph in the introduction with:

In this article, we combine a classification obtained by an HMM with the differential split-sample testing framework. The goal is to improve the robustness of the calibration/validation of a hydrological model, which is a prerequisite to climate change impact assessment. The term "robustness" refers to the ability of the hydrological model to perform well under contrasted hydro-climatic conditions. This definition is coherent with the so-called robust decision-making framework that is often advocated to handle the deep uncertainty attached to climate change (Lempert et al., 2006). This is illustrated using the Senegal River Basin (SRB) as a case study. Headwaters in the SRB are still largely natural areas (Descroix et al, 2020; Faty, 2017) and the flow regime in the upper part of the basin exhibits regime-shifting behavior with departures from the inter-annual average over extended periods of time (Faye et al. (2015); Paturel et al. (2004); Dacosta et al. (2002). These characteristics makes the SRB an interesting case study to illustrate the split-sample testing framework with hydrologic sequences identified from an HMM.

We propose replacing the abstract with:

The impacts of climate and land-use changes make the stationary assumption in hydrology obsolete. Moreover, there is still considerable uncertainty regarding the future evolution of the Earth's climate and the extent of the alteration of flow regimes. In that context, it is crucial to assess the performance of a hydrologic model over a wide range of climates and their corresponding hydrologic conditions. In this paper,
numerous, contrasted, climate sequences identified by a Hidden Markov Model (HMM) are used in a differential split-sample testing framework to assess the robustness of a hydrologic model. The split-sample test based on an HMM classification is implemented on the time series of monthly river discharges in the upper Senegal River Basin in West Africa, a region characterized by the presence of low-frequency climate signals. A comparison with the results obtained using classical rupture tests shows that the diversity of hydrologic sequences identified using the HMM can help assessing the robustness of the hydrologic model.

Section 5 only focuses on the 1940-1998 period, which displays clear subsequences for each sub-basin. To further illustrate the relevance of the HMM classification, we investigated two shorter 26-years periods (1945-1971 and 1972-1998) which were put in the Supplementary Materials section. We first decided to put those results in that section because we thought that they would somehow distract the reader from the main objective of the paper. Since both reviewers find such an analysis relevant, we will bring it back to the main text. This will give us the opportunity to discuss in more detail the importance of the length of the period.

(3) Although HMM allows for a finer labelling of each year, NSE and KGE coefficients in Table 3 do not show higher performance in fitting observed flows. Classification based on Pettitt's test provides comparable or better results, particularly for the Daka Saidou and Oualia basins. An interesting insight could be given on which is the most convenient case to apply to reach better performance (Case 1 vs Case 2; Case 3 vs Case 4;: : :.).

We would like to insist on the fact that we do not necessarily want to achieve the highest calibration/validation values. Rather, the goal is to assess the robustness, i.e. the ability of the hydrological model to perform well under contrasted hydro-climatic
conditions. Checking that a calibration on a dry state is in general better than a calibration on a wet state (or opposite) can indeed be investigated with the proposed approach but it is not the main objective of the paper. The table below nevertheless presents that comparison. It shows NSE/KGE values for calibration and validation for the seven cases.

|             |              | Pettitt Test |             | 2-states HMM |             | 3-states HMM |             |             |             |
|-------------|--------------|--------------|-------------|--------------|-------------|--------------|-------------|-------------|-------------|
| Phase       | Sub-sequence | Case 1       | Case 2      | Case 3       | Case 4      | Case 5       | Case 6      | Case 7      | Catchment   |
| Calibration | Dry          | -            | 0.935/0.967 | 0.923/0.961  | -           | 0.927/0.963  | -           | -           | Daka Saidou |
|             | Normal       | -            | -           | -            | -           | -            | 0.925/0.962 | -           |             |
|             | Wet          | 0936/0.967   | -           | -            | 0.942/0.971 | -            | -           | 0.942/0.971 |             |
| Validation  | Dry          | 0.932/0.940  | -           | -            | 0.921/0.943 | -            | 0.913/0.887 | 0.919/0.937 |             |
|             | Normal       | -            | -           | -            | -           | 0.913/0.892  | -           | 0.921/0.911 |             |
|             | Wet          | -            | 0.933/0.953 | 0.939/0.948  | -           | 0.934/0.952  | 0.938/0.920 | -           |             |
| Calibration | Dry          | -            | 0.803/0.775 | 0.775/0.864  | -           | 0.738/0.819  | -           | -           | Oualia      |
|             | Normal       | -            | -           | -            | -           | -            | 0.772/0.868 | -           |             |
|             | Wet          | 0.886/0.941  | -           | -            | 0.888/0.937 | -            | -           | 0.888/0.937 |             |
| Validation  | Dry          | 0.727/0.803  | -           | -            | 0.709/0.796 | -            | 0.717/0.786 | 0.579/0.753 |             |
|             | Normal       | -            | -           | -            | -           | 0.762/0.816  | -           | 0.714/0.795 |             |
|             | Wet          | -            | 0.836/0.845 | 0.832/0.865  | -           | 0.804/0.814  | 0.837/0.853 | -           |             |
| Calibration | Dry          | -            | 0.858/0.919 | 0.776/0.883  | -           | 0.805/0.899  | -           | -           | Bakel       |
|             | Normal       | -            | -           | -            | -           | -            | 0.827/0.909 | -           |             |
|             | Wet          | 0.900/09.41  | -           | -            | 0.899/0.949 | -            | -           | 0.913/0.943 |             |
| Validation  | Dry          | 0.693/0.533  | -           | -            | 0.715/0.678 | -            | 0.766/0.745 | 0.633/0.478 |             |
|             | Normal       | -            | -           | -            | -           | 0.782/0.734  | -           | 0.755/0.692 |             |
|             | Wet          | -            | 0.644/0.470 | 0.804/0.679  | -           | 0.622/0.436  | 0.810/0.684 | -           |             |

We can see that no case systematically stands out from the others. When calibrated over a dry sequence, validation over the wet sequence generally displays a lower criterion value. The opposite is also true. Consequently, we are not able to state that a calibration on a specific subsequence is more desirable than another one.

We will replicate this discussion for sections dealing with shorter periods (now in Supplementary materials).

(4) The case study and the results do not allow a generalization of the approach
**and probably it is not the ideal test case to show the advantages of the proposed method mostly because there is a marked rupture in streamflow observations.**

This paper shows how an HMM classification can be used within the differential split-sample testing framework to assess the robustness of hydrological models under shifting flow regimes. The Senegal River, for which the presence of low-frequency climate signals has been discussed in the literature (Ardoin-Bardin, 2004; Bodian et al, 2014, Descroix et al, 2015), is used as a case study. The results are therefore specific to the case study and cannot be generalized. However, the idea of using HMM to identify contrasted subsequences of streamflows is relevant for other rivers exhibiting regime shifting behavior. Since an HMM classification has the potential to identify more than two subsequences, the method automatically offers more flexibility to assess the robustness of a hydrological model. See our responses to your second comment for proposed changes in the manuscript.

(5) The content of the final paragraph at sect. 5.1 is not clear and differences (if any) with the traditional differential split sample test should be better outlined. Moreover, conclusions not always refer to by presented analysis and results showed in the study and do not seem to support statement in the abstract.

Both the split-sample and the differential split-sample tests belong to the calibration/validation approach proposed by Klemes (1986). Although both tests require splitting the time series, the split-sample test is applied to stationary time series whereas the differential split sample is test is used with non-stationary ones. With the split sample test, splitting the time series does not require specific modeling techniques and is mostly left to the modeler's judgment. With the differential split sample test, however, we need a technique to detect change-points in hydrological series. In this paper, change-points are detected after carrying out a HMM-based classification
of the data, which allows for the identification of multiple subsequences instead of only two with the traditional Pettit test. We have reformulated the title and rewritten the abstract to better explain the scope of the paper, which is about combining an HMM classification of hydrological series with the differential split sample test to help assessing the robustness of hydrological models.

We propose the following changes in section 5:

This article proposes an HMM-based classification to deal with complex climate sequences and shows how the resulting classification can be used in a differential split sample test to assess the robustness of a hydrological model. A modeling experiment is carried out in the Senegal River basin using the GR2M model and historical flows from 1940-1998.

The main concluding remarks are:

When records display a single point change, a classical rupture trend (as Pettitt test) remains an adequate tool to divide the records into two climate sub-sequences.

If the records contain multiple change points, an HMM classification can divide the series into several climate sub-sequences without the need for additional data. However, records must be long enough (typically 20-25 years for a 2-states HMM classification, and 30-35 years for a 3-states HMM classification).

Regardless of the division method used, the range of climate conditions over which the hydrological model can perform depends on the intrinsic variability of the series.
Compared to the Pettit test, however, the HMM classification allows for a finer labelling of the years, therefore better exploiting the intrinsic variability in the series to enrich a differential split sample test.

There are some other minor comments (see below) that I recommend to consider. Table 1: Check isohyets ranging for the Bakel sub-basin. Table 2: can be improved by associating parameters to dry, wet and nor period. Figure 5b: I suggest to use the same axis limits for both validation and calibration performances. Eq. (3): Check equation symbol (\*) Eq. (4, 5, 6): several symbols are not introduced or explained in text. Line 205: should be T3HMMnor.

Thanks for you careful reading. We will consider it in our ongoing proofreading step.

Recent examples of differential split-sample validation tests that can be included as references have been reported in:

- D.F. Motavitaab, R. Chowab, A. Guthkea, W. Nowaka. (2019). The comprehensive differential split-sample test: A stress-test for hydrological model robustness under climate variability, Journal of Hydrology, Volume 573, Pages 501-515
- H. Dakhlaouiab, D. Ruellandc, Y.Tramblayd. (2019) A bootstrap-based differential splitsample test to assess the transferability of conceptual rainfallrunoff models under past and future climate variability, Journal of Hydrology, Volume 575, Pages 470-486

Thanks for your contribution, and we will add them to the Introduction.
**References:**

Ardoin-Bardin, S. (2004). Variabilité hydroclimatique et impacts sur les ressources en eau de grands bassins hydrographiques en zone soudano-sahélienne. PhD thesis, Université Montpellier II.

Bader, J.-C., Cauchy, S., Duffar, L., and Saura, P. (2014). Monographie hydrologique du Fleuve Sénégal. De l'origine des mesures jusqu'en 2011. IRD, Marseille (France), ird edition.

Bodian, A. (2014). Caractérisation de la variabilité temporelle récente des précipitations annuelles au Sénégal (Afrique de l'Ouest). Physio-Géo, 8(Volume 8) :297-312.

Dacosta, H., Kandia, K. Y., and Malou, R. (2002). La variabilité spatio-temporelle des précipitations au Sénégal depuis un siècle. Regional Hydrology : Bridging lhe Gap between Research and Practice (Proceedings, (2) :499-506.

Descroix, L., Faty, B., Manga, S. P., Diedhiou, A. B., Lambert, L. A., Soumaré, S., Andrieu, J., Ogilvie, A., Fall, A., Mahé, G., Diallo, F. B. S., Diallo, A., Diallo, K., Albergel, J., Tanimoun, B. A., Amadou, I., Bader, J. C., Barry, A., Bodian, A., Boulvert, Y., Braquet, N., Couture, J. L., Dacosta, H., Dejacquelot, G., Diakité, M., Diallo, K., Gallese, E., Ferry, L., Konaté, L., Nnomo, B. N., Olivry, J. C., Orange, D., Sakho, Y., Sambou, S., and Vandervaere, J. P. (2020). Are the fouta djallon highlands still the water tower of west africa ?, volume 12.
Descroix, L., Diongue Niang, A., Panthou, G., Bodian, A., Sane, Y., Dacosta, H., Malam Abdou, M., Vandervaere, J.-P., and Quantin, G. (2015). Évolution récente de la pluviométrie en Afrique de l'ouest à travers deux régions : la Sénégambie et le Bassin du Niger Moyen. Climatologie, 12(Volume 12) :25-43.

Faye, C., Sow, A. A., and Ndong, J. B. (2015). Étude des sècheresses pluviométriques et hydrologiques en Afrique tropicale : caractérisation et cartographie de la sècheresse par indices dans le haut bassin du euve Sénégal.

Faty, A. (2017). Modélisation hydrologique du haut bassin versant du Fleuve Sénégal dans un contexte de variabilité hydro-climatique : Apport de la télédétection et du modèle Mike SHE. PhD thesis, Université de Cheikh Anta Diop de Dakar.

Klemes, V. (1986). Operational testing of hydrological simulation models. Hydrological Sciences Journal, 31(1) :13-24.

Lempert, R. J., Groves, D. G., Popper, S. W., and Bankes, S. C. (2006). A general, analytic method for generating robust strategies and narrative scenarios. Management Science, 52(4) :514-528.

OMVS (2011). SDAGE - Schéma directeur. Rapport téchnique.

Paturel, J.-E., Ibrehim, B., and L'Aour, A. (2004). Evolution de la pluviométrie annuelle en Afrique de l'Ouest et centrale au XXeme siècle. Sud Sciences et technologies, 13 :40.

**HESSD**

---

## Author Response (AR1)

**UNIVERSITÉ LAVAL**

**Faculté des sciences et de génie**
Département de génie civil
et de génie des eaux

12 mai. 21

Discussion paper reference: **hess-2020-596**

Dear Dr Fenicia,

Please find below the revised version of our manuscript entitle "Combining split-sample testing and Hidden Markov Modeling to assess the robustness of hydrological models". We really appreciate the feedbacks provided by the two reviewers.

We have addressed and clarified the issues raised by the reviewers. Most importantly, (1) we have changed the title so that it better reflects the scope of the manuscript; (2) as suggested by the reviewers, we have moved to the main text results that were initially in the appendix and (3) we have thoroughly revised the introduction and the material & method section so as to present first a general methodology and then its application to a suitable case study.

Detailed responses to the comments of the two reviewers are given below. We hope that you, the manuscript editor and the two reviewers are satisfied with our responses to the comments and the substantial revisions we made to the manuscript.

I look forward to hear from you again to learn whether the revised paper is acceptable for publication.

Amaury Tilmant, ing., PhD

**Point-by-point response to the reviews of Referee #1**

We highly appreciate your time and effort in conducting a thorough review of our manuscript. Please find below our responses (your comments are in bold, our responses in normal font, and proposed changes to the manuscript in italic).

**In this study, it is interesting to apply the HMM to find hidden states in a method for identifying hydro-climatic condition of data used to calibrate/validate hydrological model. It is expected that the method of identifying annual hydro-climatic states by generating hidden state sequences through HMM will be more systematic and useful. However, to improve the completeness of this paper, several supplements are needed as follows :**

**(1) Apart from the comparison between the Petitt's test and the HMM, it is necessary to present a comparative analysis of whether the climate classification sequence identified by the HMM reflects temporal variations in other meteorological data or land use. For example, it would be possible to present any changes in land use or to state whether the temporal behavior of the dry index from annual rainfall and reference evapotraspiration over the same period is similar to the sequence of climatic state identified by the HMM.**

We agree that river discharges are the result of hydrological processes taking place upstream and are influenced by changes in precipitation, land use, etc. Anthropogenic changes may indeed alter the flow regime and hence influence an HMM-derived classification which would no longer rely solely on natural factors. However, that anthropogenic influence can be removed so that the analysis is done on naturalized flows, something fairly standard in time series analysis but also in process-based hydrological modelling.

The overall goal of this paper is to highlight the relevance of combining HMMs classifications with the differential split-sample test to assess the robustness of a hydrological model to be used in climate change studies, i.e. to generate hydrologic projections from contrasted climate ones. So, the emphasis is on changes in physical processes rather than human-induced ones. The Senegal River basin is only used as a case study to illustrate the proposed method to test the robustness of a hydrological model. For that river basin, most of the runoff and headwaters of two of the three sub-basins (Daka Saidou, Oualia) are located in the Fouta Djallon, a sparsely populated plateau where vegetation cover is relatively stable (Descroix et al, 2020), anthropogenic impacts on runoff seem to be negligible (Faty, 2017) or not even mentioned in the updated Senegal River monography (Bader et al, 2014) and in the River Basin Master Plan (OMVS, 2011). The areas mainly concerned with massive land-use conversions are located downstream of Bakel, a region not considered in our analysis. For the third sub-basin, river discharges at the outlet were naturalized by Bader et al. (2014) after removing the influence of the Manantali dam on the flow regime. In this study, the term "robustness" refers to the ability of the hydrological model to perform well under contrasted hydro-climatic conditions. This definition is coherent with the so-called robust decision-making framework that is often advocated to handle the deep uncertainty attached to climate change (Lempert et al., 2006). In other words, we do not seek the "best" model (with the best fit) but a model that performs reasonably well under different conditions.

Also, we replaced the second to last paragraph in the introduction with :

*In this article, we combine a classification obtained by an HMM with the differen-*

*tial split-sample testing framework. The goal is to improve the robustness of the calibration/validation of a hydrological model, which is a prerequisite to climate change impact assessment. The term "robustness" refers to the ability of the hydrological model to perform well under contrasted hydro-climatic conditions. This definition is coherent with the so-called robust decision-making framework that is often advocated to handle the deep uncertainty attached to climate change (Lempert et al., 2006). This is illustrated using the Senegal River Basin (SRB) as a case study. Headwaters in the SRB are still largely natural areas (Descroix et al., 2020; Faty, 2017) and the flow regime in the upper part of the basin exhibits regime-shifting behavior with departures from the inter-annual average over extended periods of time (Faye et al., 2015; Paturel et al., 2004; Dacosta et al., 2002). These characteristics makes the SRB an interesting case study to illustrate the differential split-sample testing framework with hydrologic sequences identified from an HMM.*

We replaced the abstract with :

*The impacts of climate and land-use changes make the stationary assumption in hydrology obsolete. Moreover, there is still considerable uncertainty regarding the future evolution of the Earth's climate and the extent of the alteration of flow regimes. Climate change impact assessment in the water sector typically involves a modelling chain in which a hydrological model is needed to generate hydrologic projections from climate forcings. Considering the inherent uncertainty of the future climate, it is crucial to assess the performance of the hydrologic model over a wide range of climates and their corresponding hydrologic conditions. In this paper, numerous, contrasted, climate sequences identified by a Hidden Markov Model (HMM) are used in a differential split-sample testing framework to assess the robustness of a hydrologic model. The differential split-sample test based on an HMM classification is implemented on the time series of monthly river discharges in the upper Senegal River Basin in West Africa, a region characterized by the presence of low-frequency climate signals. A comparison with the results obtained using classical rupture tests shows that the diversity of hydrologic sequences identified using the HMM can help assessing the robustness of the hydrologic model.*

**(2) The three sub-basin are all located within one same basin. So their flow data show similar temporal behavior with different scale. This makes it difficult to generalize the results of this study. Moreover they show same dramatic changes in climatic conditions. This is rather thought to make it difficult to show the advantages of the proposed method in this study.**

This paper shows how an HMM classification can be used within the differential split-sample testing framework to assess the robustness of hydrological models under shifting flow regimes. The Senegal River, for which the presence of low-frequency climate signals has already been discussed in the literature (Ardoin-Bardin, 2004 ; Bodian et al, 2014, Descroix et al, 2015), is used as a case study. The results presented in this paper are therefore specific to the case study and cannot be generalized. However, the idea of using an HMM to identify contrasted subsequences of streamflows to feed a differential split sample test is relevant for other rivers exhibiting a regime shifting behavior. We modified the abstract and the introduction to better explain the novelty of the approach and the specific contributions associated with the case study (see previous comment).

**In the abstract section, the authors mentioned that the results show that when the time series of river discharges does not exhibit a clear climate trend, or when it has multiple change points, classical rupture tests are useless and HMM classification is a viable alternative as long as the climate sub-sequences**

are long enough. However, the results in section 5 do not adequately explain this. The results show that Pettitt's test is still on of the appropriate tools. Perhaps an addition of another time series (basin) should be considered that clearly illustrates the difference between methods.

In the previous version, the results section only focused on the 1940-1998 period, which displays clear subsequences for each sub-basin. To further demonstrate the relevance of the HMM classification, we analyzed two shorter 26-years periods (1945-1971 and 1972-1998) which were presented in the Supplementary Materials section. We initially decided to put that analysis in the Supplementary Materials section because we thought it might distract the attention from the main objective of the paper, which is combining differential split sample tests with HHM-based classifications. However, based on the reviewer's comment, we reconsidered that decision and moved the corresponding analysis to the main text. Also, the analysis of the three periods (1940-1998, 1945-1971 and 1972-1998) are now presented in section 3.1, 3.2 and 3.3 respectively.

**(3) The ultimate goal in hydrological modeling would be to obtain a better fit. The HMM's theoretical advantages of more granular and continuous identification is understood, but the results do not seem to support it. The authors noted in Section 5.2 that the HMM could lead to better model performances than the Pettitt's test, but it is difficult to accept the argument that the HMM is a better way with the values provided in Table 3. I think each method has similar NSE (KGE) values. A clearer rationale or explanation is needed for this part.**

The goal of this paper is not to obtain a better fit but to "assess the performance of a hydrologic model over a wide range of climates and their corresponding hydrologic conditions" (Line 5). So, the idea is not to improve the calibration of a hydrological model but to determine if the performances of that particular model measured on different, preferably contrasted, subsequences of streamflows are similar or not. That robustness is key when the hydrological model must be used in climate change studies with diverging climate projections. This point has been clarified (see our proposed changes in our response to comment 1).

**(4) The sentence for the length of data mentioned in section 6 is not the result of this paper. It was only cited from other paper and no substantive analysis was performed to support this conclusion. A minimal analysis needs to be performed to apply the claims of existing studies to the method proposed in this study.**

That sentence refers to material provided in the Supplementary Materials section of the paper. As indicated above, we brought that material back to the main text. Moreover, we added the following sentences (Line 298) :

*As $T_{1945-1971}$ is 26-years length period, the 2-states HMM derived and the 3-states HMM derived subsequences have lengths ranging from 5 to 19 years. Various authors have discussed the minimum length required for achieve a calibration or a validation without reaching a consensus, even though a number from two to eight years could be enough depending on the "hydrological events" included in the subsequences (Razavi and Coulibaly, 2013; Juston et al., 2009; Singh and Bárdossy, 2012). In our case, the technique used to identify the subsequences (HMM classifications) seeks to provide relatively homogeneous ones. Here, we assume that five years is acceptable but we do not investigate this issue*

*further since it is beyond the scope of our paper.*

**(5) Please check some typos. ex. Check the isohyets range on the Bakel basin (Table 1). ex. Check T3HMMnor in Climate segments (Figure 4).**

We particularly appreciate your careful reading, and we proofread all the paper to check for the mentioned mistakes and for any other mistakes.

**Point-by-point response to the reviews of Referee #2**

We highly appreciate your time and effort in conducting a thorough review of our manuscript. Please find below our responses (your comments are in bold, our responses in normal font, and proposed changes to the manuscript in italic).

**The study applies the differential split-sample test to evaluate the robustness of hydrological model parameters using the HMM to identify subsequences of different hydroclimatic conditions. The approach is tested over three sub-basins of the Senegal River employing the GR2M model ; the 2-states and 3-states HMM classifications are compared with the non parametric Pettitt's test which allows to detect a single change point. Authors state that results show that HMM can be a viable classification option for long time series exhibiting multiple change points.**

**There are several points that deserve further insights and that are not sufficiently explored in the study.**

**(1) The title of the paper hints at a protocol of calibration/validation of hydrological models that in my opinion is still substantially based on the differential split sample test proposed by Klemes (1986). The study rather focuses on the comparison of different approaches to identify sub-periods characterized by different climatic conditions. The 3steps protocol outlined at sec. 4 is not novel and should at least include calibration and validation steps. I suggest rephrasing the title.**

We propose the following title :

*Combining split-sample testing and Hidden Markov Modeling to assess the robustness of hydrological models.*

**(2) I agree with remark in RC1 about the necessity of a more detailed analysis to support the climate classification sequence identified by the HMM in terms of changes in meteorological data or land use.**

We agree that river discharges are the result of hydrological processes taking place upstream and are influenced by changes in precipitation, land use, etc. Anthropogenic changes may indeed alter the flow regime and hence influence an HMM-derived classification which would no longer rely solely on natural factors. However, that anthropogenic influence can be removed so that the analysis is done on naturalized flows, something fairly standard in time series analysis but also in process-based hydrological modelling.

The overall goal of this paper is to highlight the relevance of combining HMMs classifications with the differential split-sample test to assess the robustness of a hydrological model to be used in climate change studies, i.e. to generate hydrologic projections from contrasted climate ones. So, the emphasis is on changes in physical processes rather than human-induced ones. The Senegal River basin is only used as a case study to illustrate the proposed method to test the robustness of a hydrological model. For that river basin, most of the runoff and headwaters of two of the three sub-basins (Daka Saidou, Oualia) are located in the Fouta Djallon, a sparsely populated plateau where vegetation cover is relatively stable (Descroix et al, 2020), anthropogenic impacts on runoff seem to be negligible (Faty, 2017) or not even mentioned in the updated Senegal River monography (Bader et al, 2014) and in the River Basin Master Plan (OMVS, 2011). The areas mainly concerned with massive land-use conversions are located downstream of Bakel, a region not considered in our analysis. For the third sub-basin, river discharges at the outlet were naturalized by Bader et al. (2014) after removing the influence of the Manantali dam on the flow regime.

In this study, the term "robustness" refers to the ability of the hydrological model to perform well under contrasted hydro-climatic conditions. This definition is coherent with the so-called robust decision-making framework that is often advocated to handle the deep uncertainty attached to climate change (Lempert et al., 2006). In other words, we do not seek the "best" model (with the best fit) but a model that performs reasonably well under different conditions.

Also, we replaced the second to last paragraph in the introduction with :

*In this article, we combine a classification obtained by an HMM with the differential split-sample testing framework. The goal is to improve the robustness of the calibration/validation of a hydrological model, which is a prerequisite to climate change impact assessment. The term "robustness" refers to the ability of the hydrological model to perform well under contrasted hydro-climatic conditions. This definition is coherent with the so-called robust decision-making framework that is often advocated to handle the deep uncertainty attached to climate change (Lempert et al., 2006). This is illustrated using the Senegal River Basin (SRB) as a case study. Headwaters in the SRB are still largely natural areas (Descroix et al., 2020; Faty, 2017) and the flow regime in the upper part of the basin exhibits regime-shifting behavior with departures from the inter-annual average over extended periods of time (Faye et al., 2015; Paturel et al., 2004; Dacosta et al., 2002). These characteristics makes the SRB an interesting case study to illustrate the differential split-sample testing framework with hydrologic sequences identified from an HMM.*

We replaced the abstract with :

*The impacts of climate and land-use changes make the stationary assumption in hydrology obsolete. Moreover, there is still considerable uncertainty regarding the future evolution of the Earth's climate and the extent of the alteration of flow regimes. Climate change impact assessment in the water sector typically involves a modelling chain in which a hydrological model is needed to generate hydrologic projections from climate forcings. Considering the inherent uncertainty of the future climate, it is crucial to assess the performance of the hydrologic model over a wide range of climates and their corresponding hydrologic conditions. In this paper, numerous, contrasted, climate sequences identified by a Hidden Markov Model (HMM) are used in a differential split-sample testing framework to assess the robustness of a hydrologic model. The differential split-sample test based on an HMM classification is implemented on the time series of monthly river discharges in*

*the upper Senegal River Basin in West Africa, a region characterized by the presence of low-frequency climate signals. A comparison with the results obtained using classical rupture tests shows that the diversity of hydrologic sequences identified using the HMM can help assessing the robustness of the hydrologic model.*

In the previous version, the results section only focused on the 1940-1998 period, which displays clear subsequences for each sub-basin. To further demonstrate the relevance of the HMM classification, we analyzed two shorter 26-years periods (1945-1971 and 1972-1998) which were presented in the Supplementary Materials section. We initially decided to put that analysis in the Supplementary Materials section because we thought it might distract the attention from the main objective of the paper, which is combining differential split sample tests with HHM-based classifications. However, based on the reviewer's comment, we reconsidered that decision and moved the corresponding analysis to the main text. Also, the analysis of the three periods (1940-1998, 1945-1971 and 1972-1998) are now presented in section 3.1, 3.2 and 3.3 respectively.

**(3) Although HMM allows for a finer labelling of each year, NSE and KGE coefficients in Table 3 do not show higher performance in fitting observed flows. Classification based on Pettitt's test provides comparable or better results, particularly for the Daka Saidou and Oualia basins. An interesting insight could be given on which is the most convenient case to apply to reach better performance (Case 1 vs Case 2 ; Case 3 vs Case 4 ; : : :.).**

We would like to insist on the fact that we do not necessarily want to achieve the highest calibration/validation values. Rather, the goal is to assess the robustness, i.e. the ability of the hydrological model to perform well under contrasted hydro-climatic conditions. Checking that a calibration on a dry state is in general better than a calibration on a wet state (or opposite) can indeed be investigated. Also, we present three tables (Tables 3,5 and 7 respectively in sections 3.1,3.2 and 3.3) which give all calibration/validation NSE and KGE values. We can see that no case systematically stands out from the others. When calibrated over a dry sequence, validation over the wet sequence generally displays a lower criterion value. The opposite is also true. Consequently, we are not able to state that a calibration on a specific subsequence is more desirable than another one. We replicated this discussion for sections dealing with shorter periods (now in sections 3.2 and 3.3).

**(4) The case study and the results do not allow a generalization of the approach and probably it is not the ideal test case to show the advantages of the proposed method mostly because there is a marked rupture in streamflow observations.**

This paper shows how an HMM classification can be used within the differential split-sample testing framework to assess the robustness of hydrological models under shifting flow regimes. The Senegal River, for which the presence of low-frequency climate signals has been discussed in the literature (Ardoin-Bardin, 2004; Bodian, 2014; Descroix et al., 2015), is used as a case study. The results are therefore specific to the case study and cannot be generalized. However, the idea of using HMM to identify contrasted subsequences of streamflows is relevant for other rivers exhibiting regime shifting behavior. Since an HMM classification has the potential to identify more than two subsequences, the method automatically offers more flexibility to assess the robustness of a hydrological model. See our responses to your second comment for proposed changes in the manuscript.

**(5) The content of the final paragraph at sect. 5.1 is not clear and dif-**

**ferences (if any) with the traditional differential split sample test should be better outlined. Moreover, conclusions not always refer to by presented analysis and results showed in the study and do not seem to support statement in the abstract.**

Both the split-sample and the differential split-sample tests belong to the calibration/validation approach proposed by Klemes (1986). Although both tests require splitting the time series, the split-sample test is applied to stationary time series whereas the differential split sample is test is used with non-stationary ones. With the split sample test, splitting the time series does not require specific modeling techniques and is mostly left to the modeler's judgment. With the differential split sample test, however, we need a technique to detect change-points in hydrological series. In this paper, change-points are detected after carrying out a HMM-based classification of the data, which allows for the identification of multiple subsequences instead of only two with the traditional Pettit test. We have reformulated the title and rewritten the abstract to better explain the scope of the paper, which is about combining an HMM classification of hydrological series with the differential split sample test to help assessing the robustness of hydrological models.

We propose the following changes in the conclusion section (Line 336) :

*This article proposes an HMM-based classification to deal with complex climate sequences and shows how the resulting classification can be used in a differential split sample test to assess the robustness of a hydrological model. A modeling experiment is carried out in the Senegal River basin using the GR2M model and historical flows from 1940-1998. Then, two other periods have been investigated, a wet episode (1945-1971) and a dry one (1972-1998).*

*The main concluding remarks are :*

*When records display a single point change, a classical rupture trend (as Pettitt test) remains an adequate tool to divide the records into two climate subsequences.*

*If the records contain multiple change points, an HMM classification can divide the series into several climate subsequences without the need for additional data.*

*Regardless of the division method used, the range of climate conditions over which the hydrological model can perform depends on the intrinsic variability of the series. Compared to the Pettitt test, however, the HMM classification allows for a finer labelling of the years, therefore better exploiting the intrinsic variability in the series to enrich a differential split sample test.*

*We encourage the modellers to explore as many cases as possible to calibrate/validate a hydrological model according to the differential split-sample test. The parameter's stability over contrasted hydro-climatic conditions seems to depend on the studied period, on the objective functions, on the subsequences identification techniques, and the basin.*

**There are some other minor comments (see below) that I recommend to consider. Table 1 : Check isohyets ranging for the Bakel sub-basin. Table 2 : can be improved by associating parameters to dry, wet and nor period. Figure 5b : I suggest to use the same axis limits for both validation and calibration performances. Eq. (3) : Check equation symbol (*) Eq. (4, 5, 6) : several symbols are not introduced or explained in text. Line 205 : should be T3HMMnor .**

Thanks for you careful reading. We considered it.

**Recent examples of differential split-sample validation tests that can be included as references have been reported in :**

— **D.F. Motavitaab, R. Chowab, A. Guthkea, W. Nowaka. (2019). The comprehensive differential split-sample test : A stress-test for hydrological model robustness under climate variability, Journal of Hydrology, Volume 573, Pages 501-515**
— **H. Dakhlaouiab, D. Ruellandc, Y.Tramblayd. (2019) A bootstrap-based differential splitsample test to assess the transferability of conceptual rainfall-runoff models under past and future climate variability, Journal of Hydrology, Volume 575, Pages 470-486**

Thanks for your contribution, and we added them to the Introduction.

**Références**

Ardoin-Bardin, S. (2004). *Variabilité hydroclimatique et impacts sur les ressources en eau de grands bassins hydrographiques en zone soudano-sahélienne.* PhD thesis, Université Montpellier II.

Bodian, A. (2014). Caractérisation de la variabilité temporelle récente des précipitations annuelles au Sénégal (Afrique de l'Ouest). *Physio-Géo*, 8(Volume 8) :297–312.

Dacosta, H., Kandia, K. Y., and Malou, R. (2002). La variabilité spatio-temporelle des précipitations au Sénégal depuis un siècle. *Regional Hydrology : Bridging Ihe Gap between Research and Practice (Proceedings*, (2) :499–506.

Descroix, L., Diongue Niang, A., Panthou, G., Bodian, A., Sane, Y., Dacosta, H., Malam Abdou, M., Vandervaere, J.-P., and Quantin, G. (2015). Évolution récente de la pluviométrie en Afrique de l'ouest à travers deux régions : la Sénégambie et le Bassin du Niger Moyen. *Climatologie*, 12(Volume 12) :25–43.

Descroix, L., Faty, B., Manga, S. P., Diedhiou, A. B., Lambert, L. A., Soumaré, S., Andrieu, J., Ogilvie, A., Fall, A., Mahé, G., Diallo, F. B. S., Diallo, A., Diallo, K., Albergel, J., Tanimoun, B. A., Amadou, I., Bader, J. C., Barry, A., Bodian, A., Boulvert, Y., Braquet, N., Couture, J. L., Dacosta, H., Dejacquelot, G., Diakité, M., Diallo, K., Gallese, E., Ferry, L., Konaté, L., Nnomo, B. N., Olivry, J. C., Orange, D., Sakho, Y., Sambou, S., and Vandervaere, J. P. (2020). *Are the fouta djallon highlands still the water tower of west africa ?*, volume 12.

Faty, A. (2017). *Modélisation hydrologique du haut bassin versant du fleuve Sénégal dans un contexte de variabilité hydro-climatique : Apport de la télédétection et du modèle Mike SHE.* PhD thesis, Université de Cheikh Anta Diop de Dakar.

Faye, C., Diop, E. H. S., and Mbaye, I. (2015). Impacts des changements de climat et des aménagements sur les ressources en eau du fleuve sénégal : Caractérisation et évolution des régimes hydrologiques de sous-bassins versants naturels et aménagés. *Belgeo - Revue belge de géographie*, 4 :1–25.

Juston, J., Seibert, J., and Johansson, P. (2009). Temporal sampling strategies and uncertainty in calibrating a conceptual hydrological model for a small boreal catchment. *Hydrological Processes*, 23 :3093–3109.

Klemes, V. (1986). Operational testing of hydrological simulation models. *Hydrological Sciences Journal*, 31(1) :13–24.

Lempert, R. J., Groves, D. G., Popper, S. W., and Bankes, S. C. (2006). A general, analytic method for generating robust strategies and narrative scenarios. *Management Science*, 52(4) :514–528.

Paturel, J.-E., Ibrehim, B., and L'Aour, A. (2004). Evolution de la pluviométrie annuelle en Afrique de l'Ouest et centrale au XXeme siècle. *Sud Sciences et technologies*, 13 :40.

Razavi, T. and Coulibaly, P. (2013). Streamflow prediction in ungauged basins : Review of regionalization methods. *Journal of Hydrologic Engineering*, 18(8) :958–975.

Singh, S. K. and Bárdossy, A. (2012). Calibration of hydrological models on hydrologically unusual events. *Advances in Water Resources*, 38 :81–91.